# UNIFYING REGULARISATION METHODS FOR CONTINUAL LEARNING

## ABSTRACT

Continual Learning addresses the challenge of learning a number of different distributions sequentially. The goal of maintaining knowledge of earlier distributions without re-accessing them starkly conflicts with standard SGD training for artificial neural networks. An influential method to tackle this are so-called regularisation approaches. They measure the importance of each parameter for modelling a given distribution and subsequently protect important parameters from large changes. In the literature, three ways to measure parameter importance have been put forward and they have inspired a large body of follow-up work. Here, we present strong theoretical and empirical evidence that these three methods, Elastic Weight Consolidation (EWC), Synaptic Intelligence (SI) and Memory Aware Synapses (MAS), are all related to the Fisher Information. Only EWC intentionally relies on the Fisher, while the other two methods stem from rather different motivations. We find that for SI the relation to the Fisher – and in fact its performance – is due to a previously unknown bias. Altogether, this unifies a body of regularisation approaches. It provides better explanations for the effectiveness of SI- and MAS-based algorithms and offers more justified versions of these algorithms. From a practical viewpoint, our insights uncover conditions needed for different algorithms to work and allow predicting their relative performance in some situations as well as offering improvements.

## 1 INTRODUCTION

Despite considerable progress, many gaps between biological and machine intelligence remain. Animals, for example, flexibly learn new tasks throughout their lives, while at the same time maintaining robust memories of previous knowledge. This ability conflicts with traditional training procedures for artificial neural networks, which overwrite previous skills when optimizing new tasks (McCloskey & Cohen, 1989; Goodfellow et al., 2013). The field of continual learning is dedicated to mitigate this crucial shortcoming of machine learning. It exposes neural networks to a sequence of distinct tasks. During training new tasks, the algorithm is not allowed to revisit old data, but should retain previous skills while remaining flexible enough to also acquire new knowledge.

An influential line of work to approach this challenge was introduced by Kirkpatrick et al. (2017), who proposed the first regularisation-algorithm, Elastic Weight Consolidation (EWC). After training a task, EWC measures the importance of each parameter and introduces an auxiliary loss penalising large changes in important parameters. Naturally, this raises the question of how to measure this 'importance'. While EWC uses the diagonal Fisher Information, two alternatives have been proposed: Synaptic Intelligence (SI, Zenke et al. (2017)) aims to attribute the decrease in loss during training to individual parameters and Memory Aware Synapses (MAS, Aljundi et al. (2018)) introduces a heuristic measure of output sensitivity. Together, these three approaches have inspired many further regularisation-based approaches, including combinations of them (Chaudhry et al., 2018), refinements (Huszár, 2018; Ritter et al., 2018; Chaudhry et al., 2018; Yin et al., 2020), extensions (Schwarz et al., 2018; Liu et al., 2018; Park et al., 2019; Lee et al., 2020) and applications in different continual learning settings (Aljundi et al., 2019) as well as different domains of machine learning (Lan et al., 2019). Almost every new continual learning method compares to at least one of EWC, SI and MAS.

Despite their popularity and influence, basic practical and theoretical questions regarding these algorithms had previously been unanswered. Notably, it was unkown how similar these importance

measures are. Additionally, for SI and MAS (and their follow-ups) there was no solid theoretical understanding of their effectiveness. Here, we close both these gaps through a theoretical analysis confirmed by a series of carefully designed experiments on standard continual learning benchmarks. Our main findings can be summarised as follows:

(a) We show that MAS is almost identical to the 'Absolute Fisher', a quantity similar to the Fisher Information.

(b) We show that SI's importance approximation is biased, that the bias is responsible for SI's performance and that the bias, like MAS, is tightly linked to the Absolute Fisher.

(c) Our fine-grained analysis leads to new baselines, including more justified versions of MAS and SI.

(d) Together, (a) and (b) show that all three regularisation approaches (and their follow-ups) are closely linked to the Fisher Information. This unifies a large body of regularisation literature. It also gives a more solid theoretical justification for the effectiveness of SI- and MAS-based algorithms.

(e) Our precise understanding of SI allows predicting and improving its performance in different situations and offers a better performing alternative.

## 2 RELATED WORK

The problem of catastrophic forgetting in neural networks has been studied for many decades (McCloskey & Cohen, 1989; Ratcliff, 1990; French, 1999). In the context of deep learning, it received more attention again (Goodfellow et al., 2013; Srivastava et al., 2013). All previous work we are aware of proposes new or modified algorithms to tackle continual learning. No attention has been directed towards understanding or unifying existing methods and we hope that our work will not remain the only effort of this kind. We now review the broad body of continual learning algorithms. Following (Parisi et al., 2019), they are often categorised into regularisation-, replay- and architectural approaches. Most *regularisation methods* will be reviewed more closely in the next section. There we cover regularisation for standard neural nets, but the same idea has also been applied to bayesian neural networks (Nguyen et al., 2017; Swaroop et al., 2019). An alternative approach (Mirzadeh et al., 2020) does not penalize deviating from important parameters, but rather modifies training so that the network is robust to such deviations. *Replay methods* refer to algorithms which either store a small sample or generate data of old distributions and use this data while training on new methods (Rebuffi et al., 2017; Lopez-Paz & Ranzato, 2017; Shin et al., 2017; Kemker & Kanan, 2017). These approaches can be seen as investigating how far standard i.i.d.-training can be relaxed towards the (highly non-i.i.d.) continual learning setting without losing too much performance. They are interesting, but usually circumvent the original motivation of continual learning to maintain knowledge *without* accessing old distributions. Intriguingly, the most effective way to use old data appears to be simply replaying it, i.e. mimicking training with i.i.d. batches sampled from all tasks simultaneously (Chaudhry et al., 2019). *Architectural methods* extend the network as new tasks arrive (Fernando et al., 2017; Li et al., 2019; Schwarz et al., 2018; Golkar et al., 2019; von Oswald et al., 2019). This can be seen as a study of how old parts of the network can be effectively used to solve new tasks and touches upon transfer learning. Typically, it avoids the challenge of integrating new knowledge into an existing networks. Finally, van de Ven & Tolias (2019); Hsu et al. (2018); Farquhar & Gal (2018) point out that different continual learning scenarios and assumptions with varying difficulty were used across the literature.[1]

## 3 REVIEW OF EXISTING REGULARISATION METHODS

### 3.1 FORMAL DESCRIPTION OF CONTINUAL LEARNING

In continual learning we are given $K$ datasets $\mathcal{D}_1, \ldots, \mathcal{D}_K$ sequentially. When training a neural net with $N$ parameters $\theta \in \mathbb{R}^N$ on dataset $\mathcal{D}_k$, we have no access to the previously seen datasets $\mathcal{D}_{1:k-1}$. However, at test time the algorithms is tested on all $K$ tasks, usually the average accuracy is taken as measure of the algorithms performance, but see also Chaudhry et al. (2018).

---

[1]In the appendix, we critically review and question some of their experimental results.

### 3.2 COMMON FRAMEWORK FOR REGULARISATION METHODS

Regularisation based approaches introduced in Kirkpatrick et al. (2017); Huszár (2018) protect previous memories by modifying the loss function $L_k$ related to dataset $\mathcal{D}_k$. Let us denote the parameters obtained after finishing training on task $k$ by $\theta^{(k)}$ and let $\omega^{(k)} \in \mathbb{R}^N$ be the parameters' *importances*. When training on task $k$, regularisation methods use the loss

$$\tilde{L}_k = L_k + c \cdot \sum_{i=1}^{N} \omega_i^{(k-1)} \left( \theta_i - \theta_i^{(k-1)} \right)^2$$

where $c > 0$ is a hyperparameter. The first term $L_k$ is the standard (e.g. cross entropy) loss of task $k$. The second term ensures that the parameters do not move away too far from their previous values. Usually, $\omega_i^{(k)} = \omega_i^{(k-1)} + \omega_i$, where $\omega_i$ is the importance for the most recently learned task $k$.

### 3.3 REVIEW OF EWC, MAS, SI AND FOLLOW-UPS

**Elastic Weight Consolidation** (Kirkpatrick et al., 2017) uses the diagonal of the Fisher Information as importance measure. It is evaluated at the end of training a given task. To define the *Fisher Information Matrix $F$*, we omit the task index $k$ and simply consider a dataset $\mathcal{X}$. For each datapoint $X \in \mathcal{X}$ the network predicts a probability distribution $\mathbf{q}_X$ over the set of labels $L$, where we suppress dependence of $q_X$ on $\theta$ for simplicity of notation. We denote the predicted probability of class $y$ by $\mathbf{q}_X(y)$ for each $y \in L$. Let us assume that we minimize the negative log-likelihood $\log \mathbf{q}_X(y)$ and write $g(X, y) = -\frac{\partial \log \mathbf{q}_X(y)}{\partial \theta}$ for its gradient. Then the Fisher Information $F$ is given by

$$F = \mathbb{E}_{X \sim \mathcal{X}} \mathbb{E}_{y \sim \mathbf{q}_X} \left[ g(X, y) g(X, y)^T \right] = \mathbb{E}_{X \sim \mathcal{X}} \sum_{y \in L} \mathbf{q}_X(y) \cdot g(X, y) g(X, y)^T \tag{1}$$

Taking only the **diagonal Fisher** means $\omega_i(\text{EWC}) = \mathbb{E}_{X \sim \mathcal{X}} \mathbb{E}_{y \sim \mathbf{q}_X} \left[ g_i(X, y)^2 \right]$.
The *Empirical Fisher Information* replaces the expectation over $y \sim \mathbf{q}_X$ by the (deterministic) label $y$ given by the labels of the dataset. We also define the *Predicted Fisher Information* as taking the maximum-likelihood predicted label, i.e. the argmax of $\mathbf{q}_X(\cdot)$. When the network classifies images correctly and confidently, these three versions clearly become increasingly similar.
Under the assumption that the learned label distribution $\mathbf{q}_X$ is the real label distribution the Fisher equals the Hessian of the loss (Martens, 2014; Pascanu & Bengio, 2013). Its use for continual learning has a bayesian, theoretical interpretation (Kirkpatrick et al., 2017; Huszár, 2018).

**Memory Aware Synapses** (Aljundi et al., 2018) heuristically argues that the sensitivity of the function output with respect ot a parameter should be used as the parameter's importance. This sensitivity is evaluated after training a given task. The formulation in the paper suggests using the predicted probability distribution to measure parameter sensitivity. The probability distribution is the output of the neural network and the quantity that one aims to conserve during continual learning. Below we describe the precise importance this leads to. It was however brought to our attention that the MAS codebase seems to interpret the logits rather then the probability distribution as output of the neural network. In Appendix D we show that the resulting version of MAS has the same performance as the one described here, and, crucially, is also related to the Fisher Information.
Denoting, as before, the final layer of learned probabilities by $\mathbf{q}_X$, the MAS importance is implemented as (using a euclidean norm)

$$\omega_i(\text{MAS}) = \mathbb{E}_{X \sim \mathcal{X}} \left[ \left| \frac{\partial \|\mathbf{q}_X\|^2}{\partial \theta_i} \right| \right],$$

**Synaptic Intelligence** (Zenke et al., 2017) approximates the contribution of each parameter to the decrease in loss and uses this contribution as importance. To formalise the 'contribution of a parameter', let us denote the parameters at time $t$ by $\theta(t)$ and the loss by $L(t)$. If the parameters follow a smooth trajectory in parameter space, we can write the *decrease* in loss from time 0 to $T$ as

$$L(0) - L(T) = -\int_{\theta(0)}^{\theta(T)} \frac{\partial L(t)}{\partial \theta} \theta'(t) dt = -\sum_{i=1}^{N} \int_{\theta_i(0)}^{\theta_i(T)} \frac{\partial L(t)}{\partial \theta_i} \theta_i'(t) dt. \tag{2}$$

The $i$-th summand in (2) can be interpreted as the contribution of parameter $\theta_i$ to the decrease in loss. While we cannot evaluate the integral precisely, we can use a first order approximation to obtain the importances. To do so, we write $\Delta_i(t) = (\theta_i(t+1) - \theta_i(t))$ for an approximation of $\theta_i'(t)dt$ and get

$$\tilde{\omega}_i(\text{SI}) = \sum_{t=0}^{T-1} \frac{\partial L(t)}{\partial \theta_i} \cdot \Delta_i(t). \tag{3}$$

Thus, we have $\sum_i \tilde{\omega}_i(SI) \approx L(0) - L(T)$. In addition, SI rescales its importances as follows[2]

$$\omega_i(\text{SI}) = \frac{\max\{0, \tilde{\omega}_i(\text{SI})\}}{(\theta_i(T) - \theta_i(0))^2 + \xi}. \tag{4}$$

Zenke et al. (2017) use additional assumptions to justify this importance. One of the them – using full batch gradient descent – is violated in practice and we will show that this has an important effect.

**Follow up work.** We presented a version of EWC due to Huszár (2018) and tested in (Chaudhry et al., 2018; Schwarz et al., 2018). It is theoretically more sound and was shown to perform better. Chaudhry et al. (2018) combine EWC with a 'KL-rescaled'-SI. Ritter et al. (2018) use a block-diagonal (rather than diagonal) approximation of the Fisher; Yin et al. (2020) use the full Hessian matrix for small networks. Liu et al. (2018) rotate the network to diagonalise the most recent Fisher Matrix, Park et al. (2019) modify the loss of SI and Lee et al. (2020) aim to account for batch-normalisation.

## 4 MEMORY AWARE SYNAPSES (MAS) APPROXIMATES ABSOLUTE FISHER

Here, we explain why – despite its different motivation – MAS is almost linearly dependent of the **diagonal Absolute Fisher**, a variant of the diagonal Fisher taking absolute values of gradients rather than squares. We first present the theory and then carry out careful empirical confirmations of the underlying assumptions. For a summary of the definitions of algorithms and baselines see Table B.1. In Appendix D we present analogous results for MAS based on logits rather than the predicted probability distribution, showing that it performs equally and, crucially, is also related to the Fisher.

### 4.1 THEORETICAL RELATION OF MAS AND ABSOLUTE FISHER

We take a closer look at the definition of the importance of MAS. Recall that we use the predicted probability distribution of the network to measure sensitivity rather than logits. With linearity of derivatives, the chain rule and writing $y_0 = \arg\max \mathbf{q}_X$, we see (omitting expectation over $X \sim \mathcal{X}$)

$$\textbf{MAS:} \quad \left|\frac{\partial \|\mathbf{q}\|^2}{\partial \theta}\right| = 2 \left|\sum_{y \in Y} \mathbf{q}(y) \frac{\partial \mathbf{q}(y)}{\partial \theta}\right| \approx 2 \left|\mathbf{q}(y_0) \frac{\partial \mathbf{q}(y_0)}{\partial \theta}\right| = 2\mathbf{q}^2(y_0) \left|\frac{\partial \log \mathbf{q}(y_0)}{\partial \theta}\right| \tag{5}$$

Here we made the assumption that the sum is dominated by its maximum-likelihood label $y_0$, which should be the case if the network classifies its images confidently, i.e. $\mathbf{q}(y_0) \gg \mathbf{q}(y)$ for $y \neq y_0$. Recall that the importance is measured at the end of training, so that our assumption is justified if the task has been learned successfully. Using the same assumption for the Absolute Fisher we obtain

$$\textbf{Abs. Fisher:} \quad \mathbb{E}_{y \sim \mathbf{q}_X}\left[|g(X, y)|\right] \approx \mathbf{q}(y_0)|g(X, y_0)| = \mathbf{q}(y_0) \left|\frac{\partial \log \mathbf{q}(y_0)}{\partial \theta}\right|. \tag{6}$$

This shows the similarity between MAS and the Absolute Fisher. The only difference is a factor of $2\mathbf{q}_X(y_0)$, which we can assume to be approximately constant as the model learned to classify training images confidently. Even with a pessimistic guess that $\mathbf{q}_X(y_0)$ is in a range of 0.5 (rather inconfident) and 1.0 (absolutely confident), the two measures would be highly correlated.

### 4.2 EMPIRICAL RELATION OF MAS AND ABSOLUTE FISHER

Our theoretical derivation used two approximations, both of which we now explicitly check empirically on standard benchmarks; see Section 7 for a summary of our experimental setup.

---

[2]Note that the $\max(0, \cdot)$ is not part of the description in (Zenke et al., 2017). However, we needed to include it to reproduce their results. A similar mechanism can be found in the official SI code.

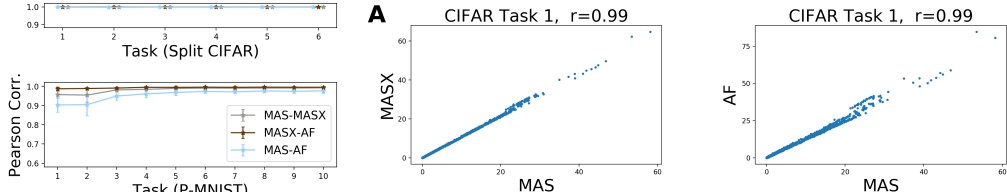

Figure 1: **Empirical Relation between MAS and baselines MASX, Absolute Fisher. Left**: Summary of Pearson Correlations (top: CIFAR, bottom: MNIST), error bars represent std over 4 runs. **Mid & Right**: Example scatter plots of importance measures. Each point in the scatter plot corresponds to one weight of the net. A straight line through the origin corresponds to two importance measures being multiples of each other as was predicted by our theoretical analysis.

**Max-Likelihood Approximation** First, in equation 5 we assumed that sum over the labels is dominated by its maximum likelihood label $y_0$. To assess the validity of this, we directly compare the entire sum (MAS) with the part only using the max-likelihood label (MASX). The results in Figure 1 show scatter plots and a summary of the pearson correlations between MAS and MASX on each task. On P-MNIST the correlations range from 0.95 to 0.99, on CIFAR they are always above 0.99. This shows that MAS and MASX are nearly equal as claimed.

**Absolute Fisher Approximation** Second, we assumed that the factor $2\mathbf{q}_X(y_0)$ distinguishing MASX (RHS of equation 5) and Absolute Fisher is roughly constant. Recall that even if $\mathbf{q}_X(y_0)$ varied between 0.5 and 1.0, the two measures would have large correlations. Figure 1 shows that we are far above this pessimistic regime with correlations between MASX and AF always above 0.98, showing that MASX approximates AF. The data also shows that the resulting correlation between MAS and AF is large (between 0.9-0.97 on MNIST, >0.99 on CIFAR), confirming that MAS approximates AF.

**Continual Learning Performance Experiment** As an additional check of our hypothesis, we implemented continual learning baselines based on MAS, MAS-MAX, AF and compared their performances. The results in Table 1(1) show that all baselines have the same performance, further supporting the claim that MAS is effective because it approximates the Absolute Fisher.

## 5 SYNAPTIC INTELLIGENCE APPROXIMATES (ONLINE) ABSOLUTE FISHER

Here, we explain why the importance of SI is related to the Absolute Fisher, despite the apparent contrast between SI's path integral and the Fisher Information. The theoretical part first identifies the bias of SI when approximating the path integral. It then relates the bias to the Absolute Fisher. As before, we carefully design baselines to validate each assumption of our analysis empirically.

### 5.1 BIAS OF SYNAPTIC INTELLIGENCE

To calculate $\omega(\text{SI})$ (equation 3), we need to calculate the product $p = \frac{\partial L(t)}{\partial \theta} \cdot \Delta(t)$ for each $t$. Evaluating the full gradient $\frac{\partial L}{\partial \theta}$ is too expensive, so SI uses a stochastic minibatch gradient. The estimate of $p$ is biased since the same minibatch is used for the update $\Delta$ and the estimate of $\partial L/\partial \theta$. We now give the calculations detailing this argument. For ease of exposition, let us assume vanilla SGD with learning rate 1 is used. Given a minibatch, denote its gradient estimate by $g + \sigma$, where $g = \partial L/\partial \theta$ denotes the full gradient and $\sigma$ the noise. The update is $\Delta = g + \sigma$. Thus, $p$ should be $p = g \cdot (g + \sigma)$. However, using $g + \sigma$, which was used for the parameter update, to estimate $g$ results in $p_{bias} = (g + \sigma)^2$. Thus, the gradient noise introduces a bias of $\mathbb{E}[\sigma^2 + \sigma g] = \mathbb{E}[\sigma^2]$.

**Unbiased SI.** Having understood the bias, we can design an unbiased estimate by using two independent minbatches to calculate $\Delta$ and estimate $g$. We get $\Delta = g + \sigma$ and an estimate $g + \sigma'$ for $g$ with independent noise $\sigma'$. We obtain $p_{no\_bias} = (g + \sigma') \cdot (g + \sigma)$ which in expectation equals $p = g \cdot (g + \sigma)$. Based on this we define an unbiased importance measure

$$\tilde{\omega}_i(\text{SIU}) = \sum_{t=0}^{T-1} (g_t + \sigma'_t) \cdot \Delta(t).$$

**Bias-Only version of SI.** To isolate the bias, we can take the difference between biased and unbiased estimate. Concretely, this gives an importance which only measures the bias of SI

$$\tilde{\omega}_i(\text{SIB}) = \sum_{t=0}^{T-1} ((g + \sigma) - (g + \sigma'_t)) \cdot \Delta(t).$$

Observe that this estimate multiplies the parameter-update $\Delta(t)$ with nothing but stochastic gradient noise. From the perspective of the SI path-integral, this should be meaningless and perform poorly. Our theory, detailed below, predicts differently.

## 5.2 RELATION OF BIAS OF SI TO ABSOLUTE FISHER

The bias of SI depends on the optimizer used. The original SI-paper (and we) uses Adam (Kingma & Ba, 2014) and we now analyse the influence of this choice in detail. We discuss the effects other optimizers, including SGD+momentum, in Appendix F, the theoretical and empirical findings are analogous to the ones presented here. Recall that $\tilde{\omega}(SI)$ is a sum over terms $\frac{\partial L(t)}{\partial \theta} \cdot \Delta(t)$, where $\Delta(t) = \theta(t+1) - \theta(t)$ is the parameter update at time $t$. Both terms, $\frac{\partial L(t)}{\partial \theta}$ as well as $\Delta(t)$, are computed using the same mini-batch. Given a stochastic gradient $g_t + \sigma_t$, Adam keeps an exponential average of the gradient $m_t = (1 - \beta_1)(g_t + \sigma_t) + \beta_1 m_{t-1}$ as well as the squared gradient $v_t = (1 - \beta_2)(g_t + \sigma_t)^2 + \beta_2 v_{t-1}$. Ignoring minor normalisations and the learning rate, the parameter update is $\Delta(t) = m_t/(\sqrt{v_t} + \epsilon)$, with $\beta_1 = 0.9$ and $\beta_2 = 0.999$. Thus,

$$\frac{\partial L(t)}{\partial \theta} \Delta(t) = \frac{(1 - \beta_1)(g_t + \sigma_t)^2}{\sqrt{v_t} + \epsilon} + \frac{\beta_1 (g_t + \sigma_t) m_{t-1}}{\sqrt{v_t} + \epsilon} \approx (1 - \beta_1) \frac{(g_t + \sigma_t)^2}{\sqrt{v_t} + \epsilon} \tag{7}$$

Here, we made **Assumption 1** that the gradient noise is larger than the gradient, or more precisely: $(1 - \beta_1)\sigma_t^2 \gg \beta_1 m_{t-1} g_t$ (we ignore $\sigma_t m_{t-1}$ since $\mathbb{E}[\sigma_t m_{t-1}] = 0$ and since we average SI over many time steps). Assumption 1 is equivalent to the bias of SI being larger than its unbiased part as detailed in Appendix G.1. Experiments in Section 5.3 provide strong support for this assumption.

Next, consider $v_t$. It is a slowly moving exponential average of $(g_t + \sigma_t)^2$ and is thus approximately $\mathbb{E}[(g_t + \sigma_t)^2]$. It is reasonable to expect that $\sqrt{v_t}$ is roughly equal to $|g_t + \sigma_t|$. This clearly does not hold for all possible distributions of $g_t + \sigma_t$, but is unlikely to be violated by non adversarially chosen, realistic noise distribution. With equation 7 and **Assumption 2** that $\sqrt{v_t} \approx |g_t + \sigma_t|$, we get

$$\tilde{\omega}(SI) = \sum_t \frac{\partial L(t)}{\partial \theta} \Delta(t) \approx (1 - \beta_1) \sum_t \frac{(g_t + \sigma_t)^2}{\sqrt{v_t}} \approx (1 - \beta_1) \sum_t |g_t + \sigma_t|. \tag{8}$$

An alternative way to arrive at equation 8 is following the analysis of Balles & Hennig (2018), who show that the update of Adam often depends on the sign of the moment $m_t$ rather than its magnitude.

**Summary.** equation 8 shows that SI approximates the *Online Absolute Fisher*, which averages the (empirical) Absolute Fisher 'online' along the training trajectory of the parameters rather than evaluating it only at the final parameter-point. We point out one subtle difference between equation 8 and AF, namely that equation 8 takes absolute values of mini-batch gradients rather than single-image gradients (like AF). In Appendix G.2 we show why this is negligible under our Assumption 1 of large gradient noise. Empirically, we also find it to be negligible, as we show below that a version of EWC based on the batched empirical Fisher has the same performance as EWC itself.

**Relation of OnAF to AF.** To see if SI, like MAS, is related to the Absolute Fisher (AF), it remains to explore the relation of OnAF and AF. If the latter two are similar, it is clear from the above that SI is similar to AF. Theoretically, in the limit of small learning rates or small Hessian (of the loss), OnAF and AF become equal. While neither of these assumption strictly hold in practice, we empirically observed - to a possibly surprising extend - strong correlations between AF and OnAF (Figure J.2). This observation still raises the question whether OnAFs relation to AF is not decreased by gradients early in training. Indeed, we present two experiments in line with this intuition in Appendix E, showing that putting more weight on later gradients can improve performance.

**Influence of Regularisation** The derivation above assumes that the parameter update $\Delta(t)$ is given by the gradients of the current task, ignoring the auxiliary regularisation loss. The latter will change update direction and momentum of the optimizer. Thus, strong regularisation will make the relation between SI and OnAF noisy, but not abolish it. This will be confirmed empirically in Section 5.3.

Table 1: **Test accuracies on Permuted-MNIST and Split CIFAR 10/100.** (Mean and std of average accuracy over 3 resp. 10 runs for MNIST resp. CIFAR) Each part compares known algorithms to baselines, explaining algorithms' performances (1-3), or showing speed-ups at equal performance (4).

| No. | Algo. | P-MNIST | CIFAR | | No. | Algo. | P-MNIST | CIFAR |
|-----|-------|---------|-------|---|-----|-------|---------|-------|
| (1) | MAS | 97.3±0.1 | 73.7±0.5 | | (3) | SI | 97.2±0.1 | 74.4±0.5 |
|     | MAS-MAX | 97.4±0.2 | 73.7±0.2 | |     | OnAF | 97.3±0.1 | 74.4±0.6 |
|     | AF | 97.4±0.1 | 73.4±0.4 | |     | AF | 97.4±0.1 | 73.4±0.4 |
| (2) | SI | 97.2±0.1 | 74.4±0.5 | | (4) | Fisher (EWC) | 97.1±0.3 | 73.1±0.7 |
|     | SI-Bias | 97.2±0.1 | 75.1±0.4 | |     | Emp. Fisher (EF) | 97.1±0.1 | 73.0±0.5 |
|     | SI-Unbiased | 96.3±0.1 | 72.5±0.8 | |     | Batch-EF | 97.1±0.1 | 73.0±0.8 |

## 5.3 EMPIRICAL INVESTIGATION OF SI, ITS BIAS AND ABSOLUTE FISHER

**Bias dominates SI & checking Assumption 1.** According to the motivation of SI the sum of importances (over parameters) $\sum_i \tilde{\omega}_i(\text{SI})$ should track the decrease in loss $L(0) - L(T)$, see equation 2. Therefore, we investigated the summed importances for SI, its unbiased version SIU and for an approximation of the path integral based on the full training set gradient (rather than approximating it). The results, Figure 2 (left) show: (1) the bias is 5-6 times larger than the unbiased part; (2) using an unbiased gradient estimate and the full gradient gives almost identical sums, supporting the validity of the unbiased estimator. Note that even the unbiased first order approximation of the path integral overestimates the decrease in loss. This is consistent with previous studies showing that the loss has positive curvature (Jastrzebski et al., 2018; Lan et al., 2019). We emphasise that the bias (difference between SI and SIU) in Figure 2 (left) is due to the term $(1 - \beta_1)\sigma_t^2$ and that this bias is considerably larger than the unbiased part. This is direct, strong evidence of Assumption 1 (see Appendix G.1 for full calculation). Additionally, a detailed analysis of the gradient noise in Appendix H gives further support for Assumption 1. Results on CIFAR are analogous (Appendix Figure J.6 ).

**Bias explains SI's performance.** We saw that the bias is much larger than the unbiased part. But how does it influence SI's performance? To quantify this, we compared SI to its unbiased version SIU and the bias-only version SIB. Note that SIB is completely independent of the path integral motivating SI, only measures gradient noise and therefore should perform poorly according to SI's original motivation. However, the results in Table 1(2), reveal the opposite: Removing the bias reduces performance of SI (SIU is worse), whereas isolating the bias does not affect or slightly improve performance (difference between SIB and SI on CIFAR is statistically significant, $p < 0.005$ two-sided t-test). This demonstrates that SI relies on its bias for its continual learning performance.

**Bias and Fisher & Checking Assumption 2.** As additional experiment, we checked how much the relation of SI to the Fisher depends on the bias. Our theory predicts that the relation between SI and OnAF is mostly based on the bias. Figure 2 (mid) confirms this, showing a strong correlation between SI, SIB and OnAF and a weaker correlation between SIU and OnAF. Note that the correlation of OnAF and SI clearly supports Assumption 2; results on CIFAR are similar (c.f. Figure J.1), but with a stronger influence of regularisation as seen in Figure 2 (right) and full text below.

**Relation of SI, OnAF to AF.** Next, we investigated the relation between OnAF and AF, finding strong correlations: between 0.73-0.87 on MNIST and 0.86-0.95 on CIFAR, see Figure J.2. Figure 2 (mid panel for MNIST, right panel for CIFAR, blue) shows that this carries over to relating SI and AF. An additional control in Appendix J.2.1 shows that this observation is not an artefact of long training times. These results confirm that SI is closely related to AF.
We clarify that here we used SI importances before the division in equation 4). We show in Appendix J that the division has close to no effect on the correlations of SI with AF.
We also show in Appendix E that putting more weight on latter summands of SI, limiting the influence of early gradients, slightly improves performance, indicating that making SI more similar to AF at the final parameter point is beneficial.

**Effect of Regularisation on SI.** Next, we assessed our derivation that large regularisation will diminish the correlation between SI and OnAF. To this end, we compared standard SI to two variants of it which have less strong regularisation. The first control simply sets regularisation strength to $c = 0$. The second control refrains from re-initialising the network weights after each task (exactly as

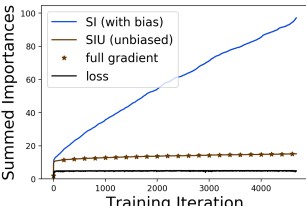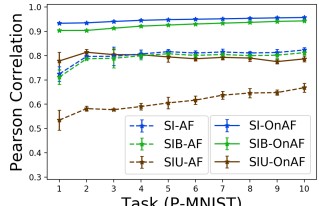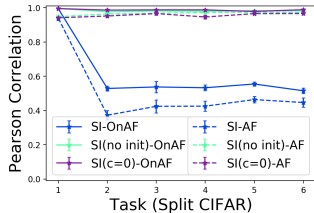

Figure 2: **Effect of Bias and Regularisation on SI. Left:** Summed Importances for SI and its unbiased version, showing that the bias dominates SI and that Assumption 1 holds. **Mid:** Pearson Correlations of SI, its bias (SIB), and unbiased version (SIU) on MNIST, showing that relation between SI and (On)AF is due to bias. **Right:** Relation between standard-SI (blue) and OnAF and two SI-controls: 'no init' (turquoise) does not re-initialise network weights after each task; '$c = 0$' (purple) has regularisation strength 0. This shows that strong regularisation weakens the tie between SI and (On)AF as explained by our theory.

in original SI, albeit with slightly worse validation performance than the version with re-initialisation). In the second setting the current parameters $\theta$ never move too far from their old value $\theta^{(k-1)}$, implying smaller gradients from the quadratic regularisation loss, and also meaning that a smaller value of $c = 0.5$ is optimal. We see that for both controls with weak regularisation the relation to OnAF is larger than for standard-SI with strong regularisation (Figure 2 (right)), consistent with our theory.

**Empirical, batched Fisher** We demonstrated that SI is related to the OnAF and pointed out that OnAF uses the Empirical Fisher and averages across batches before taking absolute values. Here, we investigate replacing the Fisher Information in EWC by the Empirical Fisher and averaging it across a mini-batch. We refer to this algorithm Batch-EF. A theoretical analysis (Appendix G.2) suggests that with moderate batch size, Batch-EF is close to EF. Empirically, we find that EF, Batch-EF and EWC have the same performance, see Table 1 (4), confirming the theoretical prediction. In this context, we note that on P-MNIST the number of samples used to approximate the Fisher and analogous quantities has some surprising effects on performance. See C.1.1 for details.

## 6 RELATION BETWEEN ABSOLUTE FISHER AND REAL FISHER

It remains to investigate the relation of Absolute and Real Fisher. Recall that (taking expectations over inputs and predicted labels) the diagonal Fisher is defined as $F = \mathbb{E}[|g + \varpi|^2]$ while the diagonal Absolute Fisher is defined as $AF = \mathbb{E}[|g + \varpi|]$, where $g + \varpi$ is the (stochastic) gradient.

Empirically, we see that AF and F are highly correlated, see Figure 3 left.

Theoretically, the exact relationship of Fisher and Absolute Fisher depends on the gradient distribution. If, for example, gradients are distributed normally $(g + \varpi) \sim \mathcal{N}(\mu, \Sigma)$ with $\Sigma_{i,i} \gg \mu_i$ (corresponding to the observation that the noise is much bigger than the gradients), we obtain $F_i \approx \Sigma_{i,i}$ and (since $|g + \varpi|$ follows a folded normal distribution) we also have $AF_i \approx c\sqrt{\Sigma_{ii}}$ with $c = \sqrt{2/\pi}$. Thus, in the case of a normal distribution with large noise, we have $F \overset{\propto}{\sim} (AF)^2$. The relationship $F \propto (AF)^2$ is in fact what we empirically observe on Split CIFAR, where correlations between $(AF)^2$ and $F$ are $> 0.98$ for all tasks and repetitions, and partly on P-MNIST, where correlations between $(AF)^2$ and $F$ are between 0.86 and 0.96.

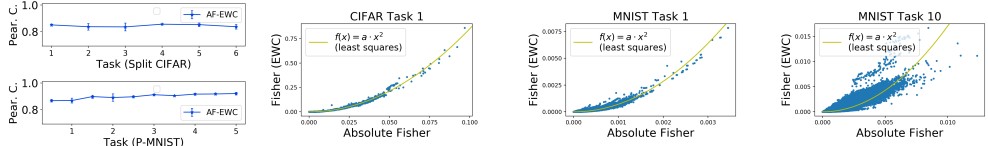

Figure 3: **Empirical Relation between Absolute Fisher and Real Fisher. Left**: Note y-axis range. Summary of Pearson Correlations (top: CIFAR, bottom: MNIST), error bars represent std over 4 runs. **Mid - Right**: Example scatter plots. Relation on Split CIFAR is nearly quadratic across tasks and repetitions. There is more variation on P-MNIST (two extremes shown). Yellow lines show least-squares fit of the model $f(x) = ax^2$ with parameter $a$.

## 7 EXPERIMENTAL SETUP

We outline the experimental setup, see Appendix C for details. We closely follow SI's setting (Zenke et al., 2017): In domain-incremental **Permuted MNIST** (Goodfellow et al., 2013) each of 10 tasks consists of a random (but fixed) pixel-permutation of MNIST and a **fully connected ReLU network** is used. The task-incremental **Split CIFAR 10/100** (Zenke et al., 2017) consists of six 10-way classification tasks, the first is CIFAR 10, and the other ones are extracted from CIFAR 100. The **keras default** CIFAR 10 **convolutional architecture** (with dropout and max-pooling) is used keras. The only difference to Zenke et al. (2017) is that like Swaroop et al. (2019), we usually re-initialise network weights after each task, observing better performance. Code is available on github. For brevity, we partly showed plots for only one benchmark, see appenidx for analogous plots.

## 8 DISCUSSION

We have investigated regularisation approaches for continual learning, which are the method of choice for continual learning without replaying old data or expanding the model. We have provided strong theoretical and experimental evidence that the three most influential methods in this domain are surprisingly similar. This does not only unify these three methods, but also their many follow ups. While the Absolute Fisher has no clear theoretical interpretation itself, its similarity to the Fisher provides a more plausible explanation of the effectiveness of MAS- and SI based algorithms.

Moreover, our algorithm SIU to approximate SI's path integral can used for the (non continual learning) algorithm LCA (Lan et al., 2019), which relies on a computationally expensive approximation of the same integral. This opens up new opportunities to apply LCA to larger models and datasets.

**Practical Relevance and Experimental Predictions.** Some may wonder what the use of understanding algorithms is. On top of the intrinsic theoretical benefit of understanding similarities between methods, our framework allows to predict in which situations algorithms will perform well and how they can be improved. Let us give one concrete example, that is relevant as distributed large batch training becomes more widespread: Our analysis reveals that SI is related to the Fisher through its bias, which is caused by gradient noise. Thus, increasing the batchsize, which decreases noise, will harm SI's performance according to our theory. We tested this specific prediction and found it to be true: With a batchsize of 2048 on P-MNIST, SI's performance degrades to $95.6 \pm 0.1$, while OnAF (with the same, large batchsize) retains $97.0 \pm 0.1$. On Split CIFAR, SI degrades to $70.0 \pm 0.9$, while OnAF retains $74.5 \pm 0.6$. We emphasise that this phenomenon is not simply explained by the change in 'training regime' (see e.g. Mirzadeh et al. (2020)), since SI, as predicted by our theory, is affected much more than OnAF by the increase in batch size. To summarise, our theory not only predicts degradation of SI correctly but also offers a well-performing alternative.

We now describe other predictions to exemplify the use of our framework and leave testing these and discovering further ones to future work. A commonly used optimisation technique is learning rate decay. Its effect on SI is that it puts more weights on those summands of SI's importance (see equation 3), which correspond to larger learning rates. Large learning rates occur early in training, thus making SI's sum less similar to the AF at the end of training.[3] Thus, we predict that learning rate decay has a negative effect on SI. In fact, we present two experiments in Appendix E showing that putting more weight on the latter summands of SI is beneficial, in line with our prediction. Interestingly, learning rate decay may be the reason that a recent empirical large-scale comparison of regularisation methods (De Lange et al., 2019) tended to observe worse performance of SI than MAS. Another somewhat puzzling observation made by De Lange et al. (2019) is that SI is more sensitive than other methods to the order in which datasetsets are presented, in cases where dataset sizes vary. In these experiments, smaller datasets are trained for fewer iterations (due to a fixed number of epochs), meaning that the sum of SI (think of OnAF) contains few summands, and is systematically smaller than for larger datasets. Thus, when first learning small datasets, SI will likely forget these since they have small importances. Starting with large datasets will not have this effect and we would expect SI to perform better. This is precisely the empirical finding in (De Lange et al., 2019).

So far, studies providing a better understanding of regularisation algorithms and their similarities had been neglected. Our contribution fills this gap and, as exemplified above, proves to be a useful tool to predict, understand and improve these algorithms.

---

[3]In the original framework of SI this is intended, while our framework predicts that it is harmful.

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

## A  APPENDIX

This appendix has several components:

1. A tabular overview over algorithms and baselines described in the paper, B
2. Full experimental details including hyperparameter seraches and values, as well as full details about Batch-EF, Appendix C.
3. Performance and Relation to Fisher of MAS based on logits, Appendix D
4. Two improvements of SI, Apprendix E
5. A more detailed account of the influence of the optimizer on SI, Apprendix F.
6. Details for two calculations omitted in the main paper, Appendix G.
7. We empirically investigate the gradient noise in Appendix H. This experiment is not specific to continual learning.
8. We critically review experimental claims from previous work about regularisation methods in Appendix I.
9. We include additional experiments and plots in Appendix J, including investigations how rescaling SI affects its relation to AF.

## B  OVERVIEW OVER ALGORITHMS AND BASELINES

Table B.1: **Summary of Regularisation Methods and Related Baselines.** *Details:* Algorithms on the left calculate importance 'online' along the parameter trajectory during training. Algorithms on the right calculate importance at the end of training a task by going through (part of) the training set again. Thus, the sum is over timesteps $t$ (left) or datapoints $X$ (right). $N$ is the number of images over which is summed. All the algorithms on the left rescale their final importances as in equation equation 4 for fair comparison. $\Delta(t) = \theta(t+1) - \theta(t)$ refers to the parameter update at time $t$, which depends on both the current task's loss and the auxiliary regularisation loss. Moreover, $(g_t + \sigma_t)$ refers to the stochastic gradient estimate of the current task's loss (where $g_t$ is the full gradient and $\sigma_t$ the noise) given to the optimizer to update parameters. In contrast, $(g_t + \sigma'_t)$ refers to an independent stochastic gradient estimate. For a datapoint $X$, $\mathbf{q}_X$ denotes the predicted label distribution and $g(X, y)$ refers to the gradient of the negative log-likelihood of $(X, y)$.

| Name | Paramater importance $\omega(\cdot)$ | | Name | Paramater importance $\omega(\cdot)$ |
|---|---|---|---|---|
| SI | $\sum_t (g_t + \sigma_t)\Delta(t)$ | | Fisher (EWC) | $\frac{1}{N} \sum_X \mathbb{E}_{y \sim \mathbf{q}_X} \left[ g(X, y)^2 \right]$ |
| SIU (SI-Unbiased) | $\sum_t (g_t + \sigma'_t)\Delta(t)$ | | AF (Absolute Fisher) | $\frac{1}{N} \sum_X \mathbb{E}_{y \sim \mathbf{q}_X} \left[ |g(X, y)| \right]$ |
| SIB (SI Bias-only) | $\sum_t (\sigma_t - \sigma'_t)\Delta(t)$ | | MAS | $\frac{1}{N} \sum_X \left| \frac{\partial \|\mathbf{q}_X\|^2}{\partial \theta} \right|$ |
| OnAF (Online AF) | $\sum_t |g_t + \sigma_t|$ | | MASX (MAS-Max) | $\frac{1}{N} \sum_X \left| \frac{\partial (\max \mathbf{q}_X)^2}{\partial \theta} \right|$ |

## C  EXPERIMENTAL DETAILS

### C.1  DETAILS, VARIANTS OF EWC, MAS AND AF.

For EWC we calculate the 'real' Fisher Information as defined in the main article.[4] For P-MNIST, we randomly sample 2000 training images (rather than iterating through the entire training set, which is prohibitively expensive). For Split CIFAR we use 500 random training images.

---

[4]We use a 'naïve' implementation to calculate the Fisher, which simply iterates over all potential labels. Theoretically, this implementation could be improved, but in practice this is infeasible since the necessary computations are currently not well-supported by standard deep-learning libraries; see Kunstner et al. (2019), Appendix B for details.

Similarly, the Empirical Fisher is based on 2000 (500) samples.

For Batch-EF, we also use 2000 (resp 500) samples and split them into 20 (resp 5) mini-batches of size 100. We compute the average gradient for each mini-batch, square it and average the squares over the mini-batches. We also multiply the result by the mini-batch size (100) for normalisation. For a theoretical justification see G.2.

MAS, MAS-X and AF algorithms are based on 1000 resp 500 random samples. When comparing these algorithms, we use the same set of samples for each algorithm.

When calculating the correlation between SI and AF, we use 30000 (MNIST) resp 500 (CIFAR) samples to evaluate AF. This is because we observed a higher variance on MNIST for AF, see Figure J.2 and subsection below.

### C.1.1 EFFECT OF SAMPLE SIZES USED TO CALCULATE FISHER

On P-MNIST, we observed for EWC that using more samples to approximate the Fisher could make performance more unstable. We believe that this is due to the following: When passing one image through a fully connected ReLU network, there will be some 'dead' neurons with zero activations. All weights connected to these neurons will have gradients equal to 0, and thus Fisher Information (for this single image) equal to 0. When more samples are used, less neurons will be 'dead' and less weights will have Fisher Information equal to 0. While this leads to a better approximation of the Fisher, changing importances to non-zero values takes away flexibility of the network for latter tasks. Empirically, this flexibility seems to help performance on P-MNIST.
We explicitly checked how the number of weights with 0 importance depends on the number of samples used: With 1000, 2000, 4000 resp 8000 samples the number of zero importances after the first task was approximately 2M, 1.8M, 1.5M resp 1.3M (out of a total 5.6M weights). We confirmed that this result is consistent across random seeds.

We note that with convolutional nets, this phenomenon of zero-importance is much less likely to appear since each weight is applied to many different neurons, which are unlikely to all be 'dead'. In line with this, we found that AF varies much less with sample size on the CIFAR task (with convolutional net) than MNIST (with fully connected net), see green line in J.2.

The fact that for EWC, MAS(X), AF there are many weights with 0 importance is likely the reason that for these methods the optimal hyperparametrisations do not re-initialise network weights after each task (see Table C.2), unlike for all other settings
If the importance of a weight is zero, re-initialising it, will move it far from its original value with no reason to return. If instead, it is not re-initialised it is more likely to not move as far away during training consecutive tasks (even if there's no direct incentive from the loss function to avoid this).

### C.2 BENCHMARKS

In **Permuted MNIST** (Goodfellow et al., 2013) each task consists of predicting the label of a random (but fixed) pixel-permutation of MNIST. We use a total of 10 tasks. As noted, we use a domain incremental setting, i.e. a single output head shared by all tasks.

In **Split CIFAR10/100** the network first has to classify CIFAR10 and then is successively given groups of 10 (consecutive) classes of CIFAR100. As in (Zenke et al., 2017), we use 6 tasks in total. Preliminary experiments on the maximum of 11 tasks showed very little difference. We use a task-incremental setting, i.e. each task has its output head and task-identity is known during training and testing.

### C.3 PRE-PROCESSING

Following Zenke et al. (2017), we normalise all pixel values to be in the interval $[0, 1]$ (i.e. we divide by 255) for MNIST and CIFAR datasets and use no data augmentation. We point out that pre-processing can affect performance, e.g. differences in pre-processing are part of the reason why results for SI on Permuted-MNIST in (Hsu et al., 2018) are considerably worse than reported in (Zenke et al., 2017) (the other two reasons being an unusually small learning rate of $10^{-4}$ for Adam in (Hsu et al., 2018) and a slightly smaller architecture).

## C.4  ARCHITECTURES AND INITIALIZATION

For P-MNIST we use a fully connected net with ReLu activations and two hidden layers of 2000 units each. For Split CIFAR10/100 we use the default Keras CNN for CIFAR 10 with 4 convolutional layers and two fully connected layers, dropout (Srivastava et al., 2014) and maxpooling, see Table C.1. We use Glorot-uniform (Glorot & Bengio, 2010) initialization for both architectures.

## C.5  OPTIMIZATION

We use the Adam Optimizer (Kingma & Ba, 2014) with tensorflow (Abadi et al., 2016) default settings ($lr = 0.001, \beta_1 = 0.9, \beta_2 = 0.999, \epsilon = 10^{-8}$) and batchsize 256 for both tasks like Zenke et al. (2017). On Permuted-MNIST we train each task for 20 epochs, on Split CIFAR we train each task for 60 epochs, also following Zenke et al. (2017). We reset the optimizer variables (momentum and such) after each task.

For the two experiments confirming our prediction regarding the effect of batchsize on SI, we made two changes on top of using a batch size 0f 2048 (and, naturally, re-tuned hyperparameters): On P-MNIST we increased the learning rate by a factor of 8, since the ratio of batchsize and learning rate is thought to influence the flatness of minima (e.g. Jastrzębski et al. (2017); Mirzadeh et al. (2020)). We did not tune or try other learning rates. On CIFAR, we tried the same learning rate adjustment but found that it led to divergence and resorted back to the default. We observed that with a batch size of 2048, already on task 2, the network did not converge during 60 epochs (for tasks 2-6, one epoch corresponds to only 5 parameter updates with this batch size). We therefore increased the number of epochs to 600 on tasks 2-6, to match the number of parameter updates of Task 1 (CIFAR10).

## C.6  SI & ONAF DETAILS

Recall that we applied the operation $\max(0, \cdot)$ (i.e. a ReLU activation) to the importance measure of each individual task (equation 4 of main paper), before adding it to the overall importance. In the original SI implementation, this seems to be replaced by applying the same operation to the overall importances (after adding potentially negative values from a given task). No description of either of these operations is given in the SI paper. In light of our findings, our version seems more justified.

Somewhat naturally, the gradient $\frac{\partial L}{\partial \theta}$ usually refers to the cross-entropy loss of the current task and not the total loss including regularisation. For CIFAR we evaluated this gradient without dropout (but the parameter update was of course evaluated with dropout).[5]

For OnAF, similarly to SI, we used the gradient evaluated without dropout on the cross-entropy loss of the current task for our importance measure.

When evaluated on benchmarks, SI, SIU, SIB and OnAF are all rescaled according to equation 4 to make the comparison as fair as possible. Scatter plots and correlations use values before this division, unless noted otherwise.

## C.7  HYPERPARAMETERS

For all methods and benchmarks, we performed grid searches for the hyperparameter $c$ and over the choice whether or not to re-initialise model variables after each task.

The grid of $c$ included values $a \cdot 10^i$ where $a \in \{1, 2, 5\}$ and $i$ was chosen in a suitable range (if a hyperparameter close to a boundary of our grid performed best, we extended the grid).

For CIFAR, we measured validation set performance based on at least three repetitions for good hyperparameters. We then picked the best HP and ran 10 repetitions on the test set. For MNIST, we measured HP-quality on the test set based on at least 3 runs for good HPs.

Additionally, SI and consequently SIU, SIB as well as OnAF have rescaled importance (c.f. equation equation 4 from main paper). The damping term $\xi$ in this rescaling was set to 0.1 for MNIST and to 0.001 for CIFAR following Zenke et al. (2017) without further HP search.

---

[5]We did not check the original SI code for what's done there and the paper does not describe which version is used.

Table C.1: CIFAR 10/100 architecture. Following Zenke et al. (2017) we use the keras default architecture for CIFAR 10. Below, 'Filt.' refers to the number of filters of a convolutional layer, or respectively the number of neurons in a fully connected layer. 'Drop.' refers to the dropout rate. 'Non-Lin.' refers to the type of non-linearity used.
Table reproduced and slightly adapted from (Zenke et al., 2017).

| Layer | Kernel | Stride | Filt. | Drop. | Non-Lin. |
|---|---|---|---|---|---|
| 3x32x32 input | | | | | |
| Convolution | 3x3 | 1x1 | 32 | | ReLU |
| Convolution | 3x3 | 1x1 | 32 | | ReLU |
| MaxPool | 2x2 | 2x2 | | 0.25 | |
| Convolution | 3x3 | 1x1 | 64 | | ReLU |
| Convolution | 3x3 | 1x1 | 64 | | ReLU |
| MaxPool | 2x2 | 2x2 | | 0.25 | |
| FC | | | 512 | 0.5 | ReLU |
| Task 1: FC | | | 10 | | softmax |
| ...:FC | | | 10 | | softmax |
| Task 6: FC | | | 10 | | softmax |

Table C.2: Hyperparameter values for our experiments. See also maintext.

| Algorithm | MNIST | | CIFAR | |
|---|---|---|---|---|
| | c | re-init | c | re-init |
| SI | 0.2 | √ | 5.0 | √ |
| SIU | 2.0 | √ | 2.0 | × |
| SIB | 0.5 | √ | 5.0 | √ |
| OnAF | 5e-5 | √ | 5e-4 | √ |
| MAS | 500 | × | 200 | √ |
| MASX | 200 | × | 1e3 | √ |
| EWC | 1e3 | × | 5e4 | √ |
| AF | 200 | × | 1e3 | √ |
| Emp. Fisher | 1e6 | × | 2e7 | √ |
| Batch-EF | 5e5 | × | 2e7 | √ |

All results shown are based on the same hyperparameters obtained – individually for each method – as described above. They can be found in Table C.2. We note that the difference between the HPs for MAS and MASX might seem to contradict our claims that the two measures are almost identical (they should require the same $c$ in this case), but this is most likely due to similar performance for different HPs and random fluctuations. For example on CIFAR, MAS had almost identical validation performance for $c = 200$ (best for MAS) and $c = 1000$ (best for MASX) ($74.2 \pm 0.7$ vs $74.1 \pm 0.5$).

Also for the other methods, we observed that usually there were two or more HP-configurations which performed very similarly. The precise 'best' values as found by the HP search and reported in Table C.2 are therefore subject to random fluctuations in the grid search.

### C.8 SCATTER PLOT DATA COLLECTION AND INTENSITY PLOTS

The scatter plots are based on a run of SI, where we in parallel to evaluating the SI importance measure also evaluated all other importance measures. Figures 1 and 2 show data based on 4 repetitions of this experiment and confirm that the correlations observed in the scatter plots are representative, it also shows the effect of sample sizes on AF.

**Intensity Plots** (as used for some figures in the appendix) Show same kind of data as scatter plots, but each weight is binned into one of 80x50 equispaced bins. The number of weights per bin is divided by the maximum number of weights per bin in that column and the resulting value is shown on a gray scale. Normalisation per column was performed since for 'global normalisation' (divide by total number of weights) only weights in the bottom left are visible (as most weights have small importances); the number of bins was chosen to match aspect ratio of plots.

### C.9 COMPUTING INFRASTRUCTURE

We ran experiments on one NVIDIA GeForce GTX 980 and on one NVIDIA GeForce GTX 1080 Ti GPU. We used tensorflow 1.4 for all experiments.

## D MAS BASED ON LOGITS

As pointed out in 3.3, there are two versions of MAS. Here, we describe results related to the version based on logits. We note that this version has the potentially undesirable feature of not being invariant to reparametrisations: If, for example, we add a constant $c$ to all the logits, this does not change predictions (or the training process), but it does change the (logit-based) MAS importance.

We found identical performance as for the other version of MAS (of course after re-tuning HPs): $97.2 \pm 0.1$ on P-MNIST and $73.9 \pm 0.7$ on CIFAR for logit based MAS, as compared to $97.3 \pm 0.1$ and $73.7 \pm 0.5$ for the version based on the probability-distribution output. For both benchmarks, the difference based on 10 repetitions was not statistically significant ($p > 0.4$, t-test).

Next, we investigated the relation between logit based MAS, MASX and the Absolute Fisher. On CIFAR we found correlations very close to 1 for all pairs. On MNIST, the correlations between MAS and MASX (based on logits) to AF are weaker than the ones for the version of MAS based on the predicted distribution. Nevertheless, they are similar to the correlation of SI and AF we observed on MNIST and above the correlations between SI and AF we observed on CIFAR. The results are shown in Figure D.1.

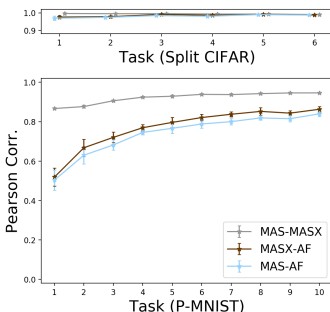

Figure D.1: **Empirical Relation between MAS and baselines MASX, Absolute Fisher.** Same as Figure 1, left, but using MAS based on logits rather than output distribution.

## E IMPROVEMENTS OF SI

As we have argued in the discussion, we expect SI to be a more faithful approximation of AF if more weight is put on the summands of SI (see equation 3), which occur later in training. As a consequence, putting more weight on latter summands could lead to better performance. We tested this prediction by running two variants of SI. For the first one, we only considered summands occuring in the second half of training (epochs 10 to 20, of 20). For the second one, we used an exponential moving average with a deacy of $0.999$. Both variants achieved an average accuracy on P-MNIST of $97.5\%$, which is better than SI's $97.2\%$. This confirms our prediction.

## F INFLUENCE OF OPTIMIZER ON SI

As pointed out in the main paper, the bias of SI depends on the optimizer used. For SGD+momentum, one can follow precisely the same steps as for Adam (simply ignoring the division by its second moment estimate) to see that in this case SI will resemble the Online Fisher, i.e. (assuming a learning rate of 1.0)

$$\tilde{\omega}(SI) \approx (1 - \beta_1) \sum_t (g_t + \sigma_t)^2$$

Analogously to Adam, in this case the relation of SI to the Fisher is due to the bias of SI. Thus, checking whether in this situation, too, the unbiased version SIU performs worse than SI, is another validation of our theory. Indeed, on P-MINST, we found that SI with SGD and momentum achieves an average of $97.4\%$, while the best run for SIU was at $96.1\%$. We did not test SGD further, due to instabilities described now: For both SI, SIU using SGD+momentum resulted in unstable behaviour, with runs occasionally diverging. This is likely due to the fact the the regularisation loss is ill-conditioned, since the importances span several orders of magnitudes. This suggests that, without further tricks and efforts, adaptive optimisers are a more suitable choice for regularisation methods.

For other ootmisers, the bias of SI will take different shapes and how this affects performance will have to be seen. Our theory suggests, that biasing SI to (a version of) the Fisher will help its performance, while we predict that fundamentally different biases lead to SI having worse performance than e.g. OnAF. An instructive example in this case is considering an approximation of natural gradient descent, which preconditions with the inverse diagonal of the empirical fisher (rather than the inverse fisher, which is expensive to compute). In this case, SI would have constant importances (across parameters) and would likely perform badly.

## G   DETAILED CALCULATIONS FOR THE BIAS OF SI, AND FOR CALCULATING FISHER IN BATCHES

Here we give the details for two calculations omitted in the main text.

### G.1   FORMULA OF BIAS OF SI WITH ADAM

Here, we show that the bias, i.e. the expected difference between SI and SIU, is equal to $(1 - \beta_1)\sigma_t^2/(\sqrt{v_t} + \epsilon)$ and that Assumption 1 is equivalent to the bias being larger than the unbiased part of SI.

Recall that for SI, we approximate $\frac{\partial L(t)}{\partial \theta}$ by $g_t + \sigma_t$, which is the same gradient estimate given to Adam. So we get

$$\textbf{SI:} \qquad \frac{\partial L(t)}{\partial \theta}\Delta(t) = \frac{(1 - \beta_1)(g_t + \sigma_t)^2}{\sqrt{v_t} + \epsilon} + \frac{\beta_1(g_t + \sigma_t)m_{t-1}}{\sqrt{v_t} + \epsilon}.$$

For SIU, we use an independent mini-batch estimate $g_t + \sigma_t'$ for $\frac{\partial L(t)}{\partial \theta}$ and therefore obtain

$$\textbf{SIU:} \qquad \frac{\partial L(t)}{\partial \theta}\Delta(t) = \frac{(1 - \beta_1)(g_t + \sigma_t')(g_t + \sigma_t)}{\sqrt{v_t} + \epsilon} + \frac{\beta_1(g_t + \sigma_t')m_{t-1}}{\sqrt{v_t} + \epsilon}.$$

Taking the difference between these two and ignoring all terms which have expectation zero (note that $\mathbb{E}[\sigma_t] = \mathbb{E}[\sigma_t'] = 0$ and that $\sigma_t, \sigma_t'$ are independent of $m_{t-1}$ and $g_t$) gives

$$\textbf{SI} - \textbf{SIU:} \qquad (1 - \beta_1)\frac{\sigma_t^2}{\sqrt{v_t} + \epsilon}$$

as claimed.

To see that Assumption 1 is equivalent to the bias being larger than the unbiased part, note that in expectation SIU equals

$$(1 - \beta_1)\frac{g_t^2}{\sqrt{v_t} + \epsilon} + \beta_1\frac{g_t m_{t-1}}{\sqrt{v_t} + \epsilon} \approx \beta_1\frac{m_{t-1}g_t}{\sqrt{v_t} + \epsilon}$$

The last approximation here is valid because $\beta_1 m_{t-1} \gg (1 - \beta_1)g_t$ which holds since (1) $\beta_1 \gg (1 - \beta_1)$ and (2) $\mathbb{E}[|m_{t-1}|] \geq \mathbb{E}[|g_t|]$. (1) follows from $\beta_1 = 0.9$, and (2) holds since $\mathbb{E}[m_{t-1}] \approx g_t$ and since $m_{t-1}$ has a noise component (due to gradient noise), while $g_t$ is a deterministic quantity, implying $\mathbb{E}[|m_{t-1}|] \geq \mathbb{E}[|g_t|]$ by elementary considerations.

In particular, the two calculations for the bias and the expectation of SIU above imply that Assumption 1, which is $(1 - \beta_1)\sigma_t^2 \gg \beta_1 m_{t-1}g_t$, is equivalent to the bias of SI being larger than its unbiased part.

## G.2 Effect of Calculating Fisher in Mini-Batches: Analysis of Batch-EF

For this subsection, let us slightly change notation and denote the images by $X_1, \ldots, X_D$ and the gradients (with respect to their labels and the cross entropy loss) by $g + \sigma_1, \ldots g + \sigma_D$. Here, again $g$ is the overall training set gradient and $\sigma_i$ is the noise (i.e. $\sum_{i=1}^D \sigma_i = 0$). Then the Empirical Fisher is given by

$$\text{EF} = \frac{1}{D} \sum_{i=1}^D (g + \sigma_i)^2$$

We want to compare this to evaluating the squared gradient over a batch. Let $i_1, \ldots, i_b$ denote uniformly random, independent indices from $\{1, \ldots, D\}$, so that $X_{i_1}, \ldots, X_{i_b}$ is a random mini-batch of size $b$. Let $g + \sigma$ be the gradient on this mini-batch. We then have, taking expectations over the random indices,

$$
\begin{aligned}
\mathbb{E}[(g + \sigma)^2] &= \mathbb{E}\left[ \frac{1}{b^2} \sum_{r,s=1}^b (g + \sigma_{i_r})(g + \sigma_{i_s}) \right] \\
&= \frac{b(b-1)}{b^2} \mathbb{E}[(g + \sigma_{i_1})(g + \sigma_{i_2})] + \frac{b}{b^2} \mathbb{E}[(g + \sigma_{i_1})^2] \\
&= \frac{b-1}{b} g^2 + \frac{1}{b} \text{EF} \\
&\approx \frac{1}{b} \text{EF}
\end{aligned}
$$

This conclusion may seem surprising, but recall that it is only based on one assumption, namely that the gradient noise is considerably bigger than the gradient itself - an assumption for which we have collected ample evidence.

This conclusion may seem surprising, but it only relies on only two assumptions: (1) We draw a minibatch with i.i.d. elements (i.e. with replacement) , and (2) the noise is considerably bigger than the gradient.

(1) If we draw minibatch elements without replacement (on the extreme end would be drawing the full batch), then our derivation ignores the contribution of $\mathbb{E}[\sigma_{i_1} \sigma_{i_2}] < 0$. Note that as long as the batchsize is small (square root of the size of the dataset or smaller) drawing a batch with or without replacement has almost the same distribution.

(2) Concretely, we assume $\mathbb{E}[(g + \sigma_i)^2] \gg (b - 1)g^2$. Note that unlike in other sections of this manuscript, here $\sigma_i$ refers to the gradient noise of individual images (rather than mini-batch noise). We have seen that even with a batch-size of 256 (i.e. when we reduce then noise by a factor of 256) the noise is still much bigger than the gradient (recall Figures J.5, J.6, H.1, H.2), so that our assumption is true with room to spare.

We also point out that the relation between the (diagonal of the) covariance matrix of minibatch gradients and the Fisher Information has also been described in several other places in the literature, e.g. Roux et al. (2008); Thomas et al. (2020); Jastrzębski et al. (2017).

## H Gradient Noise

Here, we quantitatively assess the noise magnitude outside the continual learning context. Recall that Figure 2 (left) from the main paper, as well as Figures J.5 and J.6 already show that the noise dominates the SI importance measure, which indicates that the noise is considerably larger than the gradient itself.

To obtain an assessment independent of the SI continual learning importance measure, we trained our network on MNIST as described before, i.e. a ReLu network with 2 hidden layers of 2000 units each, trained for 20 epochs with batch size 256 and default Adam settings. At each training iteration, on top of calculating the stochastic mini-batch gradient used for optimization, we also computed the full gradient on the entire training set and computed the noise – which refers to the squared $\ell_2$ distance between the stochastic mini-batch gradient and the full gradient – as well as the ratio between noise and gradient, measured as the ratio of squared $\ell_2$ norms. The results are shown in Figure H.1. In

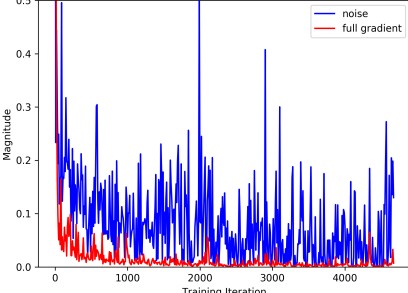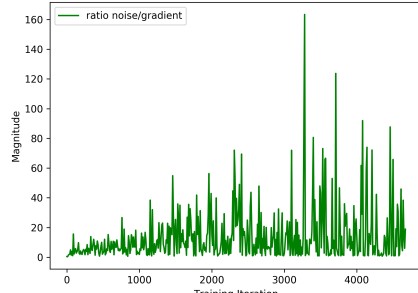

Figure H.1: Gradient noise, measured as squared $\ell_2$ distance between full training set gradient and the stochastic mini-batch gradient with a batch size of 256. 'Full gradient' magnitude is also measured as squared $\ell_2$ norm.
Data obtained by training a ReLu network with 2 hidden layers of 2000 hidden units for 20 epochs with default Adam settings. Only every 20-th datapoint shown for better visualisation.

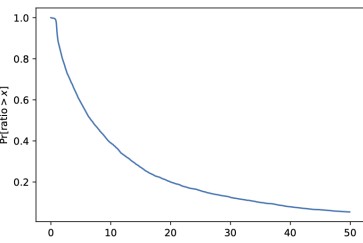

Figure H.2: $y$-value shows fraction of training iterations in which the ratio between mini-batch noise and full training set gradient was at least $x$-value. Data obtained as described in main text / Figure H.1, in particular the batch-size was 256. 'Ratio' refers to the ratio of squared $\ell_2$ norms of the respective values.

addition, we computed the fraction of iterations in which the ratio between noise and squared gradient norm is above a certain threshold, see Figure H.2.

## I    RELATIVE PERFORMANCE OF REGULARISATION METHODS

Several papers (van de Ven & Tolias, 2019; Hsu et al., 2018; Farquhar & Gal, 2018) played an important role in recognising that different continual learning algorithms had been evaluated in settings with varying difficulties. These papers also greatly clarified these setting, making an important contribution for future work.

Here, we want to point out that some of the results (and thus conclusions) of the experiments in (Hsu et al., 2018) (these were the only ones we investigated more closely) are specific to suboptimal settings and hyperparameters used there.

For example, it is found there that EWC, MAS, SI perform worse than an importance measure which assigns the same importance to all parameters (called 'L2 regularisation'). This is in contrast to results reported in the EWC paper (Kirkpatrick et al., 2017), who found the L2 baseline to perform worse than EWC. We also attempted to reproduce the L2 results and tried two versions of this approach, one with constant importance for all tasks and one with importance increasing linearly with the number of tasks. The latter approach worked better, but still considerably underperformed EWC, MAS, SI on Permuted-MNIST (88% vs 97%). We didn't run these experiments on CIFAR. An alternative importance measure based on the weights' magnitudes after training was better, but still could not

compete with SI, EWC, MAS (<95% vs >97%) showing that these importance measure are more use- and meaningful than naive baselines.

Additionally, in other work we found that for domain incremental settings (again, these were the only ones we investigated), results of baselines EWC and SI, but also of fine-tune (a baseline which takes no measures to prevent catastrophic forgetting) can be improved, often by more than 10% compared to previously mentioned reports (Hsu et al., 2018; van de Ven & Tolias, 2019). This also implies that the difference between well set-up regularisation approaches and for example repaly-methods is not nearly as big as previously thought. The differences between these results are explained by different factors (we explicitly tested eeach factor): hyperparameters (for example using Adam with the standard learning rate 0.001 improves performance over the learning rate 0.0001 used previously), initialisation (Glorot Uniform Initialisation works considerably better for SI than pytorch standard initialisation for linear layers, which reveals unexpected defaults upon close investigation), training duration (regularisation approaches usually notably benefit from longer training, this is reported in Swaroop et al. (2019), and we also found that performance on P-MNIST can be improved to 98% by training for 200 rather than 20 epochs; Hsu et al. (2018) use very short training times), data-preprocessing (depending on the setting different normalisations have different, non-negligible impacts).

# J    MORE EXPERIMENTS AND PLOTS

## J.1    PLOTS, DATA ANALOGOUS TO FIGURES/TABLES IN MAIN PAPER

Here, we show data for experiments already partly reported in the main paper, including both benchmarks and each task of each benchmark.

We start with the magnitude of the bias of SI (Figure 2, left panel): The observation that most of the SI importance is due to its bias is consistent across datasets and tasks, see Figures J.5 and J.6. Intriguingly, on CIFAR we find that the unbiased approximation of SI slightly underestimates the decrease in loss in the last task, suggesting that strong regularisation pushes the parameters in places, where the cross-entropy of the current task has negative curvature.

We show correlations between SI, SIB, SIU and AF, OnAF on CIFAR (analogous to Figure 2, middle) in Figure J.1. The findings are in accordance with the ones on MNIST reported in the main paper. Note that only the first task of CIFAR (i.e. CIFAR10) shows slightly different behaviour, as SIU and OnAF are more similar than on other tasks: correlation with OnAF is $0.994 \pm 0.001$ for SI; $0.991 \pm 0.001$ for SIU; $0.979 \pm 0.003$ for SIB. The similarity between SIU and OnAF in this situation is explained by the weights with largest OnAF importance: If we remove the 5% of weights which have highest OnAF importance, correlation between OnAF, SIU drops to $0.599 \pm 0.027$, but for SI resp. SIB the same procedure yields $0.977 \pm 0.002$ resp. $0.965 \pm 0.003$. This indicates that for the first task of CIFAR there is a small fraction of weights with large OnAF importances, which are dominated by the gradient rather than the noise. The remainder of weights is in accordance with our previous observations and dominated by noise, recall also Figure J.6.

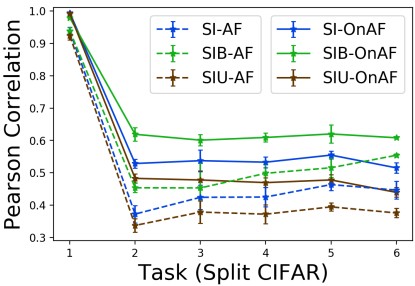

Figure J.1: **Effect of Bias on SI.** Analogous to Figure 2 (middle) but on CIFAR rather than MNIST. Drop of correlations for tasks 2-6 is due to strong correlation as predicted in Section 5.2 and confirmed in Figure 2 (right).

We did not conduct the analogue of experiment corresponding to Figure 2, right panel, on MNIST since there the influence of regularisation was already weak there.

## J.2 RELATION OF ONAF AND AF

We show summaries of the correlations between OnAF and AF in Figure J.2. This includes comparisons of AF based on different sample sizes. We also investigated the influence of training duration on AF, OnAF, see next section.

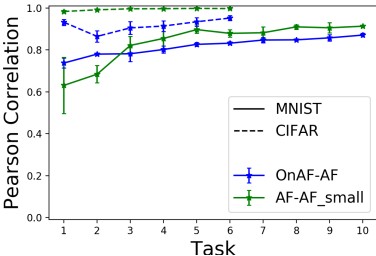

Figure J.2: **Correlations between AF and OnAF.** Solid lines show correlations on MNIST, dotted lines show correlations on CIFAR. 'AF' is based on 30000 (MNIST) resp. 5000 (CIFAR) samples, while 'AF_small' is based on 1000 resp. 500 samples.

### J.2.1 INFLUENCE OF TRAINING DURATION ON AF, ONAF

It seems plausible that the correlation between AF and OnAF we observed is due to training on tasks for a long time – the model might converge quickly so that the sum of OnAF is dominated by summands which approximate AF close to the final point in training. To test this, we trained our networks for varying numbers of epochs on the first task of P-MNIST and Split CIFAR10/100 and measured the correlation between OnAF and AF. The results in Tables J.1, J.2 show that the correlation between AF, OnAF does not rely on long training duration. On MNIST, we even see decreasing correlation with longer training time, this may be due to either an actual decrease in correlation or due to higher variance when estimating AF using 10000 samples (or both).

Table J.1: **Correlation between AF and OnAF on CIFAR10 depending on training time.** We show mean and standard deviation of Pearson correlation over 3 runs. AF is based on 500 samples. If the standard deviation is below 0.005 it is shown as 0.00.

| NUMBER OF EPOCHS | CORRELATION |
| --- | --- |
| 1 | $0.94\pm 0.01$ |
| 2 | $0.95\pm 0.00$ |
| 4 | $0.96\pm 0.01$ |
| 8 | $0.96\pm 0.01$ |
| 15 | $0.97\pm 0.00$ |
| 30 | $0.97\pm 0.00$ |
| 60 | $0.95\pm 0.01$ |

## J.3 ADDITIONAL SCATTER PLOTS AND INTENSITY PLOTS

We show additional scatter and intensity plots in Figure J.7. Note that for both architectures there are a few million weights. The scatter plots are overcrowded and show the whole range of dependencies between two measure that can possibly occur. Intensity plots show the dependencies that the majority of weights adheres to.

Table J.2: **Correlation between AF and OnAF on MNIST depending on training time.** We show mean and standard deviation of Pearson correlation over 3 runs. AF is based on 10000 samples. Recall that for 30000 samples and 20 epochs correlation is larger.

| NUMBER OF EPOCHS | CORRELATION |
|:---:|:---:|
| 1 | $0.90 \pm 0.01$ |
| 2 | $0.90 \pm 0.01$ |
| 5 | $0.88 \pm 0.01$ |
| 10 | $0.78 \pm 0.04$ |
| 20 | $0.68 \pm 0.05$ |

### J.4 ALTERNATIVE COMPARISON OF SI, SIU, SIB TO AF, ONAF

Finally, we point out that our comparisons between SI, SIU, SIB and AF, OnAF are usually obtained after applying a $\max(0, \cdot)$ (c.f. equation equation 4 of main paper) to the SI (and SIU, SIB) importances (the algorithms diverge without this operation) and before the division in equation 4 (our theory doesn't make predictions about this division).

Here, we first show that the division in equation 4 does not affect correlations between SI and AF, see Figure J.3.

Next, we present results without the max operation (and without division) in J.4, further confirming our analysis that (1) correlation between OnAF and SI is weakened by strong regularisation and (2) the correlation is due to the bias of SI.

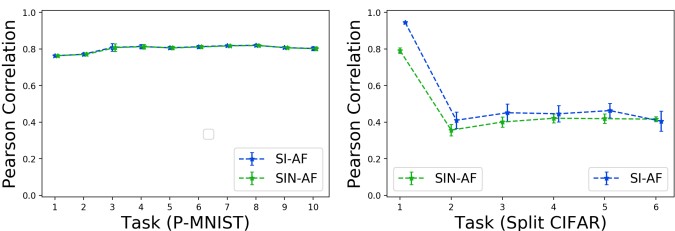

Figure J.3: Analogous to Figure 2 (middle) from main paper, comparing SI and SIN to AF. SI refers to the importance after applying the $max(\cdot, 0)$ operation, but before the division. SIN also includes the division (or **N**ormalisation), see equation 4.

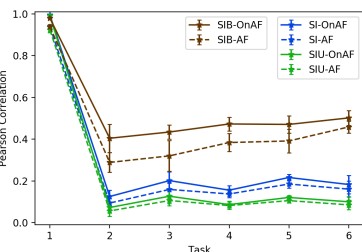

Figure J.4: Analogous to Figure 2 (middle) from main paper, comparing SI, SIU, SIB but before the operation $\max(\cdot, 0)$ is applied to importances of SI, SIU, SIB. See fulltext. Data from CIFAR tasks. On P-MNIST the operation $\max(\cdot, 0)$ has close to no effect.

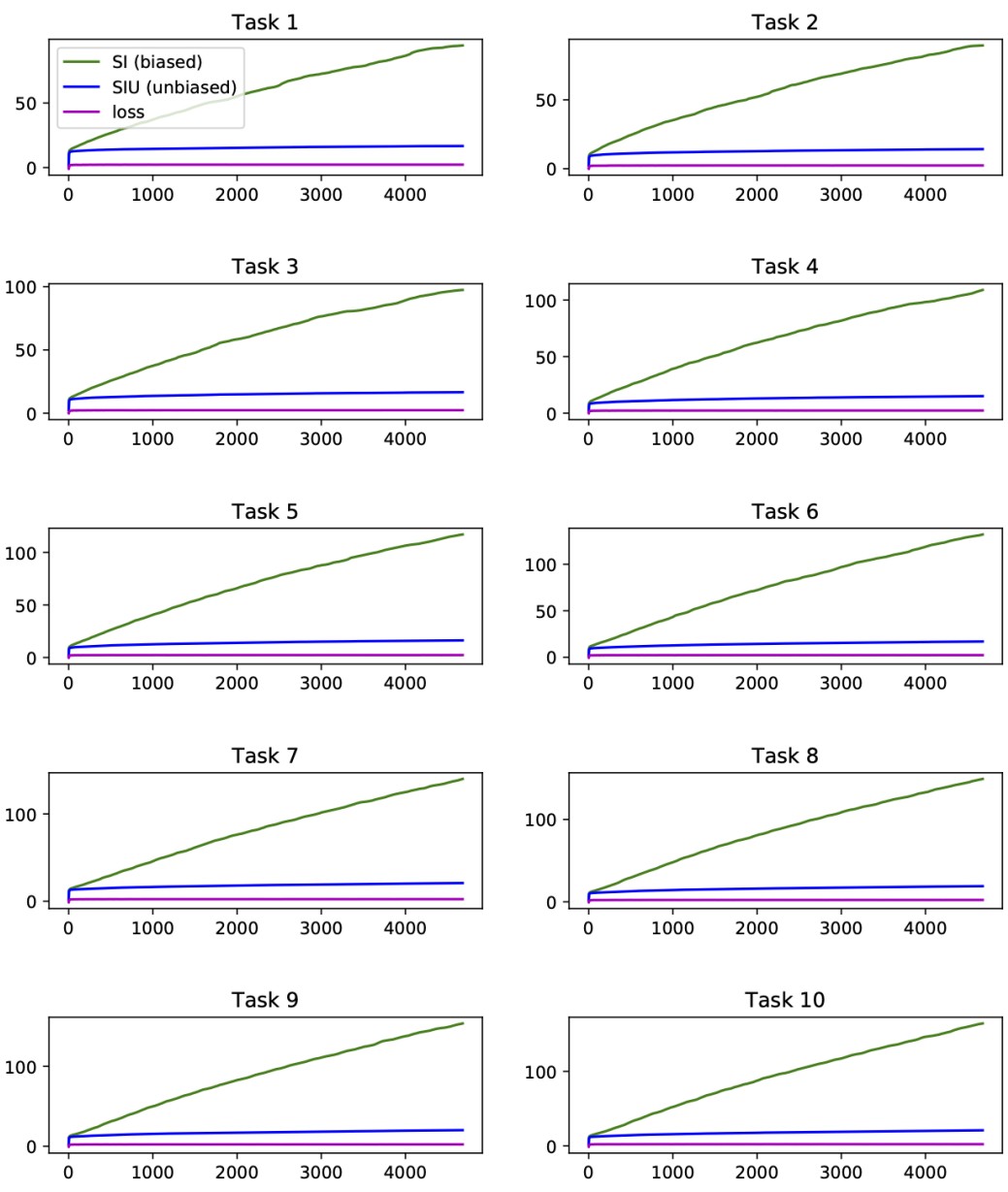

Figure J.5: Summed importances for all P-MNIST tasks for SI and its unbiased version. Analogous to Figure 2 (left) from main paper.

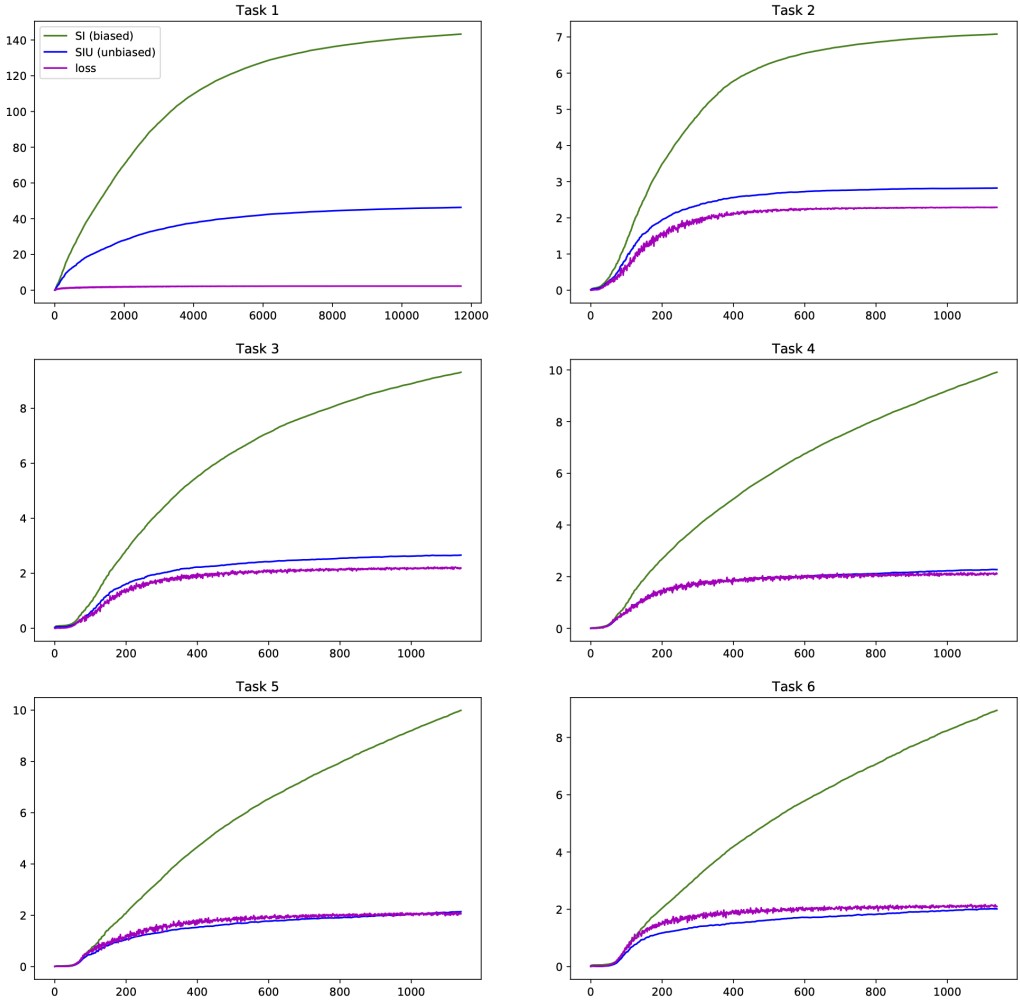

Figure J.6: Summed importances for all Split CIFAR 10/100 tasks for SI and its unbiased version. Analogous to Figure 2 (left) from main paper.

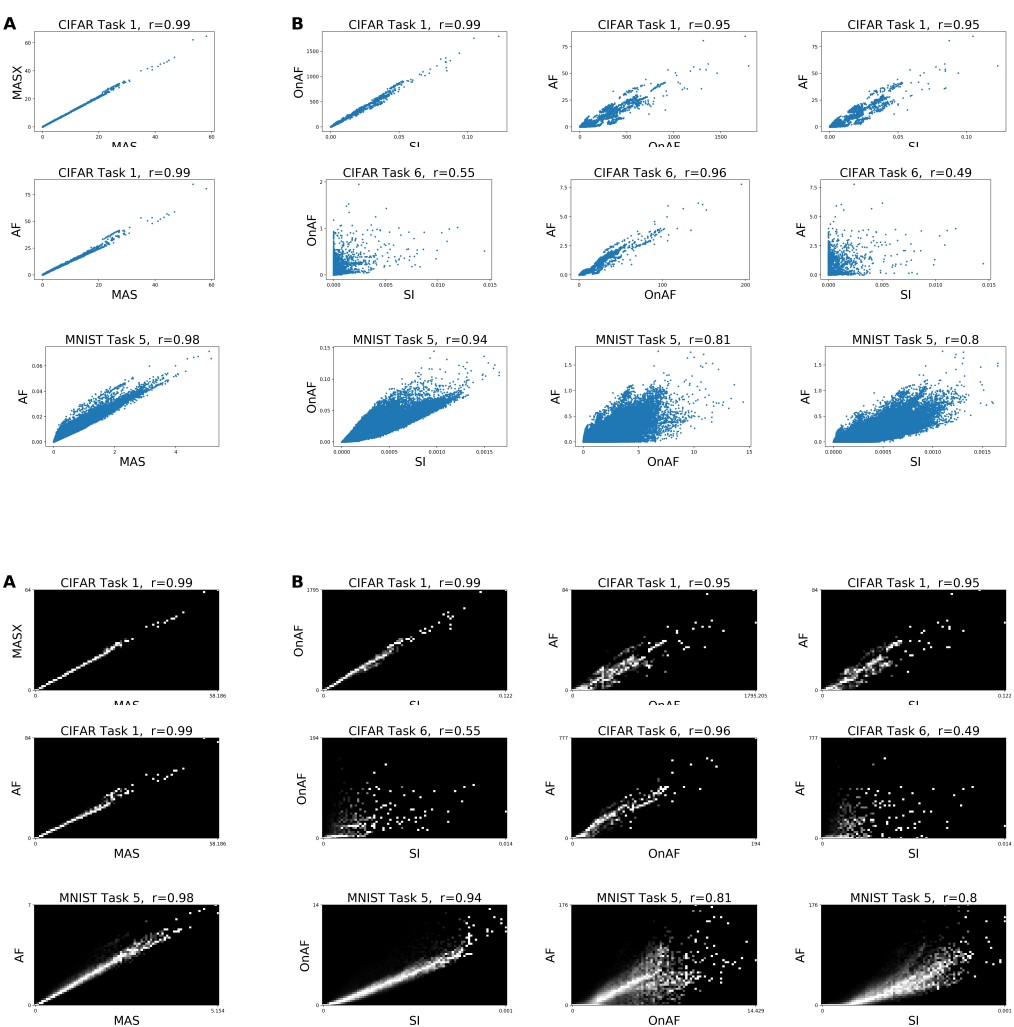

Figure J.7: More scatter and intensity plots. Refer to the title of each panel for details.

