# OpenReview forum: "Unifying Regularisation Methods for Continual Learning"
_ICLR.cc/2021/Conference — Reject_

### Official Review · AnonReviewer4 · 2020-10-25
**Interesting but some limitations**

**Rating:** 5
**Confidence:** 3

**Review:**

Disclaimer: I am not an expert in continual learning even if I have already experimented with EWC, but rather my main expertise is related to FIM in other contexts.

This works aims at unifying 3 popular regularisation type continual learning methods, namely EWC, SI and MAS, by showing that under some assumptions they all relate to the Fisher Information Matrix. These assumptions are shown to hold on 2 different tasks trained using Adam.

I however found several limitations while reading your paper:

### Limitations regarding MAS

In the github repo of MAS (it's not entirely clear from their paper), they seem to consider the function $F$ to be the score for each class (i.e. the output of the last linear transformation), while in your paper your $\mathbf q_X$ is the class probability (i.e. the softmax). So their $\omega\left(\text{MAS}\right)$ is different from yours.

### Limitations regarding SI

Your argument regarding SI (sec 5.2) seems to be valid only when using Adam, while SGD+momentum is the standard for image classification nowadays. In my opinion you can come up with a more general argument even using standard stochastic gradient algorithm, e.g. the relationship between the FIM and the covariance of the minibatch gradients as already been studied e.g. in [1] and [2]. If your empirical analysis and conclusion that it is the bias that explains SI's performance still holds with SG (without Adam), then the argument is straightforward.

Moreover the denominator $\theta\left(T\right) - \theta\left(0\right)$ of eq. 4 is not discussed later in the text, or in other words, if $\tilde\omega\left(\text{SI}\right)$ is similar to $\omega\left(\text{EWC}\right)$, what about $\omega\left(\text{SI}\right)$?

### Conclusion

In conclusion, I really appreciate the effort of trying to unify which is certainly more useful than inventing countless slightly different variants of the very same technique. I would however like to see these points addressed before publication.


[1] Le Roux, N., Manzagol, P. A., & Bengio, Y., Topmoumoute online natural gradient algorithm, NeurIPS 2008.

[2] Thomas, V., Pedregosa, F., Merriënboer, B., Manzagol, P. A., Bengio, Y., & Le Roux, N., On the interplay between noise and curvature and its effect on optimization and generalization. In International Conference on Artificial Intelligence and Statistics, AISTATS 2020

---

> ### Author Response · Authors · 2020-11-15
> **Response Rev 4**
>
> Thank you for your considerate, to-the-point review, we found that it raised several important points.
>
>
> **Limitations regarding MAS**
> That’s a very important point, thank you bringing it to our attention. Before describing how this affects results, we briefly want to point out that we re-read the MAS paper, and still think that the “output of the neural network” most naturally corresponds to the predicted probability distribution, and eventually this is what one wants to preserve during continual learning. But that’s a side note, of course.
> We implemented the alternative, logit-based version of MAS. Performance is identical to our version of MAS: 97.3 vs 97.2 and 73.7 vs 73.9, averaged across 10 runs, the difference was not statistically significant (p>0.4 on both benchmarks).
> Next, we tested how correlated logit-MAS is to the Absolute Fisher (theoretically the relation is  hard to quantify precisely without additional assumptions). On CIFAR, the correlations are above 0.98 for each task, on MNIST they are weaker than before but still similar to the correlation between SI and AF. Full details and plots are included in the updated paper: The results are summarized in the beginning of Section 4, the Figure and an extended discussion can be found in Appendix D).
> In short, both versions of MAS perform equally well and both are consistent with our hypothesis that correlation with AF explains continual learning performance.
>
>
> **Limitations regarding SI**
> You are right that the relation of SI to the Fisher depends on the choice optimizer. We also acknowledge this in the beginning of section of 5.2.
>
> *Regarding theory*
> One can see that using momentum-SGD will lead to SI strongly resembling the “Online Fisher” (the only assumption is that the gradient noise is large than the momentum term, which we’ve already seen to be true for Adam). We added this explanation, as well as another example, and the experiment described below to the paper (Appendix F, page 17; also beginning of Section 5.2 page 6), since we agree that due to its widespread use momentum-SGD should be discussed.
>
> *Experiments with SGD, rather than Adam*
> We had already run SGD-momentum based SI and we found it to agree with our predictions. SI (biased) had an average performance of 97.4%, while the best run of SIU (unbiased) achieved 96.1%. The reason we originally didn’t include this result in the paper is that both SI and SIU are unstable when trained with momentum-SGD. For both, roughly 1 in 10 runs diverged. It seems plausible that this is because of the auxiliary regularization loss, which is pretty ill-conditioned since the parameter-importances span several orders of magnitude.
>
> On a related note, we now also describe the effect of learning rate decay on SI in the updated Discussion, addressing also another optimization technique, that is common nowadays.
>
> *Normalising SI*
> Thank you also for pointing out this relevant point. We have added figures showing the correlation between the normalized version of SI  (after division) to AF (Figure J3, page 25). The correlations between SI and AF are almost unaffected by normalization. We explicitly point out the difference between the SI importances in the main paper to clarify the matter.
>
>
>
> We hope that this addresses your points regarding both MAS and SI. Either way, thank you for raising points that improved our paper.
> If you have any further concerns, please let us know.

---

### Official Review · AnonReviewer3 · 2020-10-27
**Three Methods Awkwardly Unified with Weak Logic**

**Rating:** 3
**Confidence:** 4

**Review:**

## Summary

This paper attempts to unify the three most prominent regularization-based continual learning methods: EWC, MAS, and SI.
While EWC has a solid theoretical justification under certain assumptions, the other two are based on intuition and heuristics.
The authors show that the importance weights of MAS and SI are similar to the diagonal absolute Fisher.

---

## Pros

- Connecting the path integral of SI to the Absolute Fisher is interesting.
- The paper is well-organized and easy to follow.

---

## Cons

### Lack of justification for the Absolute Fisher

The Absolute Fisher seemingly appears from nowhere. In contrast to the diagonal Fisher, it does not seem to have any other interpretation. The authors merely show that MAS and SI are similar to using Absolute Fisher as importance weights. The similarity alone is not enough to be a theoretical explanation for effectiveness. I think the authors should justify that the Absolute Fisher is an optimal regularization under certain assumptions.

### Weak connection between the diagonal Fisher Information and the Absolute Fisher

Despite searching the web, I could not find any proper material on the Absolute Fisher that explains its connection to the diagonal Fisher.
The only similarity that I find between the Absolute Fisher and the original Fisher is that they can be computed with the gradient. I do not see any reason to call it the Absolute *Fisher*.
Therefore, I cannot agree with the claim that this paper presents a unified framework, just because it fits several methods into distinct concepts that have "Fisher" common in their names.

### Doubts about the value of a unified framework

I ask the authors a more fundamental question: do we really need a unified framework? Although the derivation of EWC is mathematically grounded, it heavily relies on certain assumptions about the shape of the loss surface. The assumptions make the use of the diagonal Fisher information an optimal choice. However, considering its poor CL performance, those assumptions seemingly do not hold in deep neural networks.

Similarly, I think other methods, such as MAS and SI, can also be optimal under different assumptions. Therefore, I argue that it is more meaningful to investigate which condition/assumption makes a certain algorithm optimal, rather than framing all algorithms into one unified framework. I suspect that the latter case is not even possible.

The experiments also support my claim: AF does not necessarily outperform MAS or SI. Instead, the focus of this paper is the correlation among methods. I think the best regularization method varies depending on the specific model architecture and task design.

### Minor issues
- Assumption 1 in Section 5.2 depends heavily on the batch size. The gradient noise will quickly diminish as the batch size grows.

---

## Overall evaluation

It is hard for me to agree with this paper's fundamental motivation: a unified framework for regularization methods. Also, the overall logic of this paper is too weak. Since I could not see other utility in this paper, I recommend rejection.

---

## Post rebuttal

In response to my doubts about a unified framework, the authors claimed that their *theory* could *predict* an algorithm or hyperparameter's performance.
Since there was no description of the theory, I assumed that the theory is:
> If an algorithm is different from AF, its performance is expected to be poor.

And the authors refuted my interpretation:
> We claim that SI and MAS work because they are similar to AF.

The authors claim that the theory somehow applies to SI and MAS but not others. However, I could not find any description of why the applicability is restricted and to what extent it is applicable. I think these are vital parts of a proper theory. Without them, a theory is useless since we cannot decide whether it applies to a new CL algorithm until we actually run some tests.

Also, I want to emphasize that association is not causation. The authors should have claimed, "SI and MAS work, **and** they are similar to AF," instead of "SI and MAS work **because** they are similar to AF."

Even after the discussion with other reviewers, my concerns are not resolved.
Therefore, I retain my initial rating.

---

> ### Author Response · Authors · 2020-11-15
> **Response Rev 3**
>
> Thank you for your review. It is a shame you found our unification “awkward” and our logic “weak”.  Still, we are convinced that our contribution is valuable offering useful insights and we politely ask you to consider our rebuttal and reconsider your opinion.
>
> **RE: Doubt about Value of Findings**
> Let’s start with some common ground. When describing your doubts, you state that you would find it interesting to know under which conditions SI and MAS work well. We find that interesting, too, and actually our paper allows to make predictions of exactly this kind by uncovering conditions needed for certain algorithms to work.
> We want to give one very concrete example for this:
> In your review (“Minor Issue”), you correctly point out that our derivation of SI being related to AF is only correct for `small’ batch sizes [we quantify this in Appendix G2 (before update: D2)]. So what happens for large batch sizes (which are used increasingly often for distributed training these days)? Clearly the noise will diminish, and therefore the bias of SI. Thus, SI will be less similar to AF (as explained in the paper) and our theory predicts that it will therefore perform worse. We tested this prediction by running SI on P-MNIST with batch size 2048 and comparing it to OnAF as baseline. Si achieved 95.6% while OnAF achieved 97.0% (averaged across 3 runs, both standard deviations below 0.1%). That’s a considerable difference on this benchmark, predicted by our theory. Note also that our theory did not only predict this difference but also offered a way to avoid the performance loss by using OnAF.
> To sum up, our theory does exactly what you ask for – it predicts correctly in which settings algorithms work and don’t work. Moreover, it offers improvements in situations in which algorithms do not work.
> We have updated the Discussion to account for this. There, we also discuss additional applications of our findings pointing out relations to a recent large scale empirical investigation of regularization methods..
>
>
> **RE: Weak Connection between Fisher and Absolute Fisher**
> Let us restate the equation for the diagonal Fisher, denoting the gradient by $g$ and taking expectations over datapoints and predicted labels:
> $$F = E[g^2]$$
> Let us also restate the equation for the diagonal Absolute Fisher, using the same notation:
> $$AF = E[|g|]$$
>
> It is evident that the two are not just arbitrary functions of the gradient, but much more similar. The reason we called the latter “Absolute Fisher” is that it takes the “absolute” value $|g|$ rather than the square $|g|^2$, and that other than that is virtually identical to the  “Fisher”. [Note that all operations, square as well as absolute value, are elementwise.] This should address your concern about our choice of name.
>
> Even more importantly, we show in the paper and in the example above, that closeness to the Absolute Fisher explains the performance of regularisation methods.
> Let us re-iterate why this is the case:  Taking the path-integral motivation presented in the original SI paper more seriously by using an unbiased estimate makes SI less similar to the Absolute Fisher (as we show theoretically and empirically). It also reduces performance. In other words, similarity of SI to AF explains SI’s performance.
> Now consider the batch size. Increasing it reduces the bias of SI, and thus makes it less similar to the Absolute Fisher (as explained theoretically and empirically). It also reduces performance. In other words, performance of SI is explained by its similarity to the Absolute FIsher.
>
>
> **RE: Justification of Absolute Fisher**
> We have demonstrated above that our explanation based on the Absolute Fisher allows making predictions about the performance of algorithms.
> Leaving aside theory, which relies on assumptions, which are violated in practice (which you also point out in the context of EWC), this is the ultimate test for any explanation: Does it enable you to make correct predictions?
> We have established above that our explanation does allow precisely this.
>
>
> **RE: Minor Issue**
> This should be addressed by the points above.
>
> Lastly, we fully agree with you that the best regularization method will depend on the specific setup. We do not intend to rank the methods discussed. We simply believe (and have demonstrated!) that by understanding why certain regularization methods work, we can make informed predictions about how setups will influence performance.

---

> > ### Comment · AnonReviewer3 · 2020-11-16
> > **Re: Response Rev 3**
> >
> > Since Reviewer 1 and I share similar concerns, I also checked the response to Reviewer 1. However, I still could not find proper justification for AF. Here I clarify some points further.
> >
> > ### Justification of Absolute Fisher
> > The main argument of the authors' response is summarized as follows:
> > - If an algorithm is different from AF, its performance is expected to be poor.
> >
> > However, it requires the assumption that AF is the optimal CL algorithm (at least under certain conditions). Without this assumption, one cannot predict that an algorithm will perform worse just because it is different from AF. Nonetheless, the authors do not provide proof of this assumption. If the performance of the SI+large batch should drop simply due to its difference from AF, what about EWC? Should EWC perform poorly because EWC is different from AF?
> >
> > ### Connection between Fisher and Absolute Fisher
> > The authors' response confirms my initial argument: the only thing in common between the two is that they can be computed with the gradient. Yes, I do see the similarity between the formula $\mathbb E [g^2]$ and $\mathbb E [|g|]$. But that is not what I asked for.
> >
> > EWC uses Fisher information in the context of Laplace approximation and utilizes the equivalence between the Fisher information matrix and Hessian under some regularity conditions. Therefore, the real goal of EWC is to approximate the precision of the posterior, and the Fisher information was just a computational shortcut. In contrast, Absolute Fisher does not have such a theoretical background and seems too arbitrary. The authors seem to ignore this crucial difference.
> >
> > In the response to Reviewer 1, the authors said:
> > > EWC is only close to something that makes sense theoretically. Similarly, the Absolute Fisher is clearly close to something that makes sense. If you believe that EWC might work, it’s not unreasonable to expect the Absolute Fisher to work. Mere theory can’t make predictions about either version.
> >
> > Then, with the same logic, it is not unreasonable to expect $\mathbb E [|g|^{1.5}]$ or $\mathbb E [|g|^{0.9}]$ to work. Is there any concrete reason that $\mathbb E [|g|]$ should be better than those?

---

> > > ### Author Response · Authors · 2020-11-16
> > > **Clarifying a Misinterpretation**
> > >
> > > Thank you for taking the time to reply.
> > >
> > > Let us start by clarifying a crucial misunderstanding. You summarise our main argument as follows:
> > >
> > > > If an algorithm is different from AF, its performance is expected to be poor.
> > >
> > > But we do not claim or imply this anywhere in the paper or our reply.  We do not believe that it is true either.
> > > We claim that SI and MAS work because they are similar to AF. AF is clearly is close to the Fisher and we empirically show it to work well. But this does not imply and we do not claim that everything other than AF performs poorly.
> > > We also claim that making SI less similar to AF and more similar to the path integral makes it worse. This also does not say that everything other than AF performs poorly. It only implies that the path integral does not work as well as AF.  Importantly, this shows that AF is a better explanation for SI’s effectiveness than the motivation from SI’s original publication. This is relevant for anyone working with SI. We give concrete examples for the relevance in the Discussion.
> > >
> > > In a similar vein, it is not at all unreasonable to expect $\mathbb{E}[|g|^{0.9}]$ or $\mathbb{E}[|g|^{1.5}]$ to work. In fact, we are almost certain that they would work. They are very likely close enough to something that makes sense (the Fisher). If you like, we can run these algorithms and check that they work.
> > > Please note that the fact these two algorithms probably work does not at all stand in opposition to our claims.
> > >
> > >
> > > **Motivation of AF**
> > >
> > > We do not introduce AF as a new CL algorithm claiming that it is better than alternatives.
> > >
> > > We merely claim that AF is useful to explain SI and MAS. We make this claim because introducing AF is what uncovers the similarity of SI and MAS to the Fisher.
> > > We also present additional evidence for AF being a good explanation in the paper and in the Discussion. Theoretical optimality is not part of the evidence, but there is plenty of other evidence. In fact, the explanation turns out to be good enough to allow correct predictions, which one would not have made without it.

---

> > > > ### Comment · AnonReviewer3 · 2020-11-17
> > > > **Re: Clarifying a Misinterpretation**
> > > >
> > > > In the previous reply, I summarized the authors' response as
> > > > > If an algorithm is different from AF, its performance is expected to be poor.
> > > >
> > > > However, the authors strongly refuted this summarization. Then, how should I interpret the following sentences excerpted from the earlier response?
> > > > > So what happens for large batch sizes? Clearly the noise will diminish, and therefore the bias of SI. Thus, **SI will be less similar to AF and our theory predicts that it will therefore perform worse**.
> > > >
> > > > I was not sure what the *theory* is in this context. In the paper, I could not find any *theory* for predicting a CL algorithm's performance. Without seeing any experimental result, how can one be so sure that SI with a larger batch will perform worse, not better?
> > > >
> > > > Therefore, I had to conclude that the only plausible interpretation is
> > > > > If an algorithm is different from AF, its performance is expected to be poor.
> > > >
> > > > If I am wrong, the authors should precisely describe the *theory* that predicts a CL algorithm's performance. However, if there really is such a great theory, why is it not the main theme of the paper?

---

> > > > > ### Author Response · Authors · 2020-11-17
> > > > > **Re**
> > > > >
> > > > > In our reply we wrote:
> > > > >
> > > > > > We also claim that making SI less similar to AF and more similar to the path integral makes it worse.
> > > > >
> > > > > This is a more precise version of our claim. In the paper, this is clear from the context.
> > > > >
> > > > > Similarly, what exactly our “theory” is, is clear from the context: We precisely describe it in the sentences preceding our prediction about SI (see Discussion)

---

### Official Review · AnonReviewer1 · 2020-10-27
**Not convinced by connection through the "absolute Fisher"**

**Rating:** 5
**Confidence:** 4

**Review:**

******************************************************************
********************** POST DISCUSSION UPDATE **********************
******************************************************************
Thank you to the authors for the discussion. Given that the relationship between AF and F has now been addressed, I will increase my score. However, since the connection mainly hinges on the empirical correlation between the two, I still don't think that the paper quite establishes the theoretical connection between the three continual learning methods that it aims for. Finally, I'm not quite convinced by the potential impact of the insights, which seem fairly specific to choosing the batch size of SI, so overall I think the paper still is below the acceptance threshold.
******************************************************************
************************** END OF UPDATE***************************
******************************************************************

The paper unifies two regularisation-based continual learning methods from the literature (Synaptic Intelligence and Memory Aware Synapses) by arguing that they both approximate the "Absolute Fisher", a variant of the Fisher Information that averages absolute instead of squared gradients, to determine the regularisation weights. By this the authors claim to establish a relationship between these two approaches and Elastic Weight Consolidation, which approximates the diagonal of the Fisher.

While the paper is well-written and -structured, and SI and MAS are popular methods in the literature, I'm not convinced by the link through the "Absolute Fisher". The Fisher Information is a well-studied quantity with many appealing statistical properties and even a subtle change to its definition such as replacing the expectation over the labels with the empirical data breaks many of these (see the referenced Kunstner et al. paper). So simply taking absolute instead of squared gradients and stating that this is "a natural variant" is not a sufficient basis for a paper, especially considering that the term has not appeared anywhere in the literature before as far as I could tell. I think the authors really need to either provide some references to existing work or establish themselves that this variant makes sense theoretically. So overall my recommendation is to **reject** the paper.

Further, I'm not really sure what the takeaway from the proposed unification of these methods is even assuming that it can be put on a more solid foundation. Is it that through the relationship to EWC they are all approximately Bayesian? How would this inform future work? I feel like the paper pokes in that direction through empirically comparing some variants of SI and MAS, but most variants perform almost identically and are highly correlated, so again I am not sure what exactly to take away from this.  The part on SI relying on its 'bias' seems potentially interesting, but since -- if I understand things correctly -- this is an unexpected empirical result from the theoretical point the paper is trying to establish, it would be necessary to go a bit more in depth. To summarise, I think the authors need to more clearly explain and establish the impact of their work.

As a minor final point, I do not find the proposed efficient version of EWC convincing at all. First, estimating the Fisher as the square of the averaged gradients has been done before, for example it is implemented in the [tensorflow KFAC codebase](https://github.com/tensorflow/kfac/blob/3ee1bec8dcd851d50618cd542a8d1aff92512f7c/kfac/python/ops/fisher_factors.py#L945-L950). Second, the empirical comparison does not make much sense, since Jacobians in Pytorch can be calculated efficiently for linear and convolutional layers without too much effort through backward hooks (see e.g. [the backpack library](https://backpack.pt/) for an implementation). In more recent versions, Pytorch also provides autograd functions for computing jacobians natively -- I'm not sure how efficiently they are implemented, but in any case the empirical comparison here needs to use an efficient, batched implementation for calculating the Fisher in order for it to be meaningful. Calculating gradients for individual data points is neither sensible nor necessary.

---

> ### Author Response · Authors · 2020-11-15
> **Response Rev 1**
>
> Thank you for your review. As you may have expected, we disagree with some of your points. Concretely, we do not think  that your concerns directly affect our contributions. We explain this below and would appreciate if you considered our arguments with an open mind. Maybe this will require slightly updating what you consider the main contributions of our paper. We clarify this here and in the updated paper.
>
> **Fisher is beautiful, but does not give provable guarantees**
> We agree and are aware that the Fisher has many nice properties. But it is worth remembering that EWC relies on a series of approximations: Taking the Fisher rather than the Hessian (which is only locally valid), taking only the diagonal , and putting an unjustified hyperparameter in front of the regularization loss.
> While EWC is close to something that makes sense, its basis is already too vague to allow real theoretical guarantees. We will explain below why this is relevant.
>
> **Motivation for Absolute Fisher**
> You asked us to establish that the Absolute Fisher makes sense theoretically. We would like to emphasize several related points:
> -	EWC  is only close to something that makes sense theoretically. Similarly, the Absolute Fisher is clearly close to something that makes sense. If you believe that EWC might work, it’s not unreasonable to expect the Absolute Fisher to work. Mere theory can’t make predictions about either version.
> -	Bear in mind that we did not invent SI and MAS. In that sense it is not our fault that they don’t strictly make sense. Our contribution is giving a better explanation of their effectiveness, uncovering their similarity to the Fisher. This gives a better reason to belief that they work than we had before.  If we want to test how good our explanation is, we can use it to make predictions. If these predictions turn out to be correct, the explanation has merit. We will come back to this below.
>
> This being said, we agree that calling AF a “natural variant” is a misnomer. We updated this and other formulations in the paper.
>
>
> **SI relying on its bias is not a surprising empirical results. It is in line with our theory and a strong confirmation thereof**
>
> You pointed out that SI relying on its bias is an interesting result and raise the question whether it is just an empirical and surprising observation.
> First, we are glad that you agree that this is interesting. Second, we want to clarify our point. Our main hypothesis is that SI & MAS work because they approximate AF. We show that the bias of SI is the reason that it is related to AF. Thus, the bias should lead to better performance. That’s exactly what we find empirically.
> We’re sorry that this was not clear from our manuscript. We modified  the end of section 5.2 and if you point us to another section that may be misleading, we are happy to fix it – this result is an important part of our contribution as it is a direct confirmation of our main hypothesis.
>
> **Batch-EF**
> We accept your point that making fair wallclock time comparisons requires tuning implementations more than we did and removed claims about speed-ups. Thanks for pointing out this inaccuracy. We have replaced this part by other experiments and improvements to demonstrate direct applications of our insights, see below.
>
> **How does this inform future work?**
> Beyond the intrinsic value of giving a better explanation for existing algorithms, we want to give one very concrete example of how our results inform practitioners (we guess that’s what you’re after; for theoreticians the benefit of better understanding something should be enough). Suppose you have many GPUs and switch to distributed training with large mini-batches. This is an increasingly common scenario and presumably, one would expect one’s continual learning algorithms to not be affected by this. However, our theory predicts otherwise. We have shown that SI relies on its relation to AF, and that this relation is due to the bias caused by stochastic gradient noise. Increasing batch-size decreases gradient noise and therefore the relation of SI to AF. In short, our theory predicts that Si will lose performance with large minibatches. To test this prediction, we increased the batch size in our experiments to 2048 and compared SI to OnAF. Si achieved 95.6% while OnAF achieved 97.0% (averaged across 3 runs, both standard deviations below 0.1%). That’s a considerable difference on this benchmark, predicted by our theory. Note also that our theory did not only predict this difference but also offered a way to avoid the performance loss by using OnAF.
>
> We hope that this convinces you that understanding algorithms is useful, not only in hindsight but also for applications and future work relying on these algorithms. We also hope that it convinces you that the Absolute Fisher is a useful explanation; after all it passed the ultimate test for scientific explanations – it allowed making correct predictions.

---

> > ### Comment · AnonReviewer1 · 2020-11-16
> > **Re: Response Rev 1**
> >
> > To start, perhaps let me explicitly state the main contribution the paper aims for based on my understanding of the manuscript: to connect EWC, SI and MAS as approximating the same quantity (the Fisher) via the AF. Your rebuttal seems to argue that it is establishing the AF as a useful quantity for continual learning and possibly other settings -- which would be a perfectly fine route to go for, but that is not reflected whatsoever in the paper itself, so I will not adjust my score based on that argument.
> >
> > In my view, the main weakness remains basing the connection between the AF and Fisher on the claim that the absolute value and square are "similar". Approximating the Hessian as the Fisher makes sense, in fact the Bernstein-von-Mises theorem would suggest using the Fisher directly to obtain a Gaussian approximation to the posterior. Similarly, the diagonal approximation can make sense and has been used successfully in countless papers over the past few years, which gives it at the very least a strong empirical backing. The connection between the AF and Fisher comes completely out of the blue -- the presence of other approximations does not automatically justify yet another approximation, it needs to be established either theoretically or empirically (ideally both) in its own right.
> >
> > It is not my intention to deny the value of connecting MAS and SI through the absolute Fisher, but including EWC in this has a rather weak basis in my opinion. It may seem less appealing to unify two methods instead of three and to lose the connection to the method that has theoretical grounding in Bayesian online learning, but I would be much more inclined to accept a more narrow, but well-argued paper.
> >
> > As a final note, I would suggest not reading too much into results on permuted MNIST. That benchmark has been falling out of favor over the past couple of years (in my opinion rightfully so) for being too easy to solve, see e.g. (Farquhar & Gal, 2018). Using more challenging benchmarks will make any empirical findings more robust. Further, the drop in performance for a larger batch size would also be predicted by the work of Mirzadeh et al. (2020).
> >
> >
> > References:
> >
> > Farquhar & Gal. Towards Robust Evaluations of Continual Learning. arxiv preprint arXiv:1805.09733
> >
> > Mirzadeh et al. Understanding the Role of Training Regimes in Continual Learning. arxiv preprint arXiv:2006.06958

---

> > > ### Author Response · Authors · 2020-11-17
> > > **Re: Response Rev 1**
> > >
> > > Thank you for your reply. We appreciate that you engage in the discussion despite the disagreement.
> > >
> > > From your response we get the impression that:
> > > -	You find our insights about MAS and SI valuable (presumably also well justified).
> > > -	The main disagreement comes from how we contextualise these findings with respect to the Fisher.
> > >
> > > If this is the case, the disagreement seems about wording/presentation rather than the actual results and experiments carried out, and we are more than happy to find a consensus and address this issue. We include a few suggestions at the very bottom of this reply and we are open to more feedback.
> > >
> > >
> > > Regarding our prediction and the Mirzadeh et al. paper: There is an important difference in predictions. The Mirzadeh et al. paper predicts that all CL methods alike will suffer from large batch sizes. In contrast, our explanation predicts that SI in particular will suffer more than other methods. Our prediction is confirmed since SI is considerably worse than the baseline OnAF [Naturally, the baseline has the same, large batch-size as SI, so that the difference is not explained by Mirzadeh et al.]
> > > It is not possible to make these predictions  (see Discussion) without the understanding of SI that we gained. In particular, the original motivation of SI uses full-batch gradient descent for their theoretical analysis, suggesting - if anything - that larger batch-sizes give a more faithful approximation and should work better. Our explanation predicts that larger batch sizes work worse, and is correct. It also offers a better alternative, which does not suffer from large batch sizes.
> > >
> > > Regarding the AF approximation, you suggest that it should be established empirically or theoretically (or ideally both). We want to point out that we do establish it empirically by running the AF algorithm on CL benchmarks, the context relevant for our paper.
> > >
> > > **AF as an explanation/useful quantity**
> > > This might be an issue of presentation/wording. We want to explain what we mean by “explain” and "useful". Again, we are very open for feedback.
> > > -	Imagine someone presenting MAS. At face value, MAS preserves the euclidean norm of the prediction vector. There are many bad predictions that have the same norm as the learned prediction, which we want to preserve. So it is not clear that MAS will work. Now, if someone writes down a few lines of equations showing the similarity of MAS with the “Absolute Fisher”, which is strikingly similar to the Fisher, wouldn’t you find it much more plausible that MAS works? If the answer is ‘yes’ that’s exactly what we mean by “explain”.
> > > -	The same goes for SI. The fact that the removing the bias degrades performance appears inexplicable, unless somebody writes down a few equations showing that the bias is similar to the Fisher. This should make the effectiveness of the bias much more plausible. (As a side note, the biased SI importance becomes equal to an online sum of Batch-Empirial-Fisher if one uses SGD rather than Adam, showing a yet stronger link to the Fisher).
> > >
> > > Independently of what word is used, the “explanation” gives a new understanding of these methods. We have demonstrated that this understanding leads to correct, non-obvious predictions. This is the sense in which we think AF is a useful quantity.
> > >
> > >
> > >
> > >
> > >
> > > Some Suggestions to change presentation:
> > >
> > > We can change the title from “unifying” to “understanding”. Of course this would also mean replacing other occurrences of “unify” and instead stating that SI and MAS are similar to AF, which is theoretically unjustified but undeniably similar to the Fisher.
> > > We would not be willing to omit at least pointing out the similarity between AF and the Fisher.
> > > In abstract, intro and discussion, we can replace “similarity to AF provides an explanation” by “provides a more plausible explanation” [than we previously had].
> > > We can replace “AF is a variant of Fisher” by “AF is similar to Fisher”.
> > > and of course additional changes in the same spirit.

---

> > > > ### Comment · AnonReviewer1 · 2020-11-19
> > > > **Re: Response Rev 1**
> > > >
> > > > Thank you for your reply. I'm afraid that our disagreement goes beyond just wording and presentation. First, I strongly disagree that your experiments establish a connection between the AF and F -- apologies if I'm misinterpreting your response here, but my understanding here is that you're effectively saying that because the AF works for continual learning, it must be related to the F. This would only be the case if the F was the only reasonably quantity to use for regularization in continual learning, but of course there may be others (as you also state yourselves in the response to R3). Secondly, my answer to "wouldn’t you find it much more plausible that MAS works?" would at best be a "maybe". The reason for it not being a "yes" is that the AF **is not** the F. Yes, the definitions appear similar. But that makes their similarity a reasonable **hypothesis**. The purpose of a paper is not to present a hypothesis, but to support it with evidence.
> > > >
> > > > The similarity between the AF and F that you claim is a key component of your chain of reasoning (MAS/SI approximate AF -> AF is similar to F -> F has a principled motivation -> MAS/SI approximate a reasonable quantity and are therefore expected to work). And again, the equations merely appearing similar is just not a sufficient basis for a paper. Carefully pointing out the apparent similarity is fine, but this does not establish a connection to EWC without further evidence. Even a seemingly minor change in the definition of the Fisher -- such as for the empirical Fisher -- can break most properties of the F (Kunstner et al., 2019).
> > > >
> > > > To make this a bit more constructive, the claims that you can -- in my view -- support at this point are that MAS and SI are similar, the AF works for continual learning and that SI relies on a small batch size, which supports that the AF is what makes it work (I agree with your point on the Mirzadeh reference; I would however like to see the bigger drop in performance with large batches for SI confirmed on split-CIFAR). So I could see a paper written around the connection between MAS and SI through the AF and the AF as a potentially interesting quantity for continual learning. Pointing our the possible(!) connection EWC would be fine as an avenue to propose for further investigation. However, such a paper would differ significantly in spirit from what you have submitted, so that I don't think it would be a reasonable revision at this point for ICLR.
> > > >
> > > >
> > > > Reference:
> > > >
> > > > Kunstner et al. Limitations of the empirical fisher approximation for natural gradient descent. In NeurIPS 2019.

---

> > > > > ### Author Response · Authors · 2020-11-20
> > > > > **Re**
> > > > >
> > > > > Thank you for taking the time to clarify again, and pointing out the missing link in our chain of reasoning. We apologise for missing and misunderstanding this point earlier in the discussion.
> > > > >
> > > > > We added a section explicitly investigating the 'simialrity’ between AF and F both empirically and theoretically.
> > > > > - Theoretically, if one assumes (for example) gradients to be normally distributed $\mathcal{N}(\mu, \Sigma)$ with $\Sigma_{i,i} \gg \mu_i$ (corresponding to the observation that the noise is much bigger than the gradients), then we can see that $F \propto (AF)^2$. There are other assumptions on the distribution of gradients, that lead to the same conclusion.
> > > > > - Empirically, $F \propto (AF)^2$ this is a good approximation on both datasets, but more so on CIFAR, and explains the high correlations between AF and F (around 0.9 on Permuted-MNIST and around 0.85 on Split CIFAR). See the newly added Section 6 and Figure 3 for details.
> > > > >
> > > > > This should put the 'smilarity’ of AF to F on more solid basis.
> > > > >
> > > > > We have also updated the phrasing around the Absolute Fisher to make more clear that it does not have its own theoretical motivation but benefits from the (now more precisely investigated) similarity to the Fisher.
> > > > >
> > > > > Regarding the prediction and limitation of P-MNIST, we also confirmed the prediction on Split CIFAR: With batchsize 2048, SI drops to $70.0 (\pm 0.9)$ average accuracy, while with the same batchsize OnAF remains at $74.6 (\pm 0.5)$.

---

### Official Review · AnonReviewer2 · 2020-10-28
**AnonReviewer2 Review**

**Rating:** 6
**Confidence:** 5

**Review:**

**Summary of paper**

This paper draws links between three common regularisation methods for continual learning: EWC, MAS, and SI. It shows that MAS and SI approximate the Absolute Fisher matrix. The authors provide many experiments to test their claims and assumptions. Finally, the authors also propose a cheaper way to run EWC.

**Review summary**

I really like the majority of this paper. Unifying these regularisation methods is great, and not obvious (particularly in the case of SI). The accompanying experiments are crucial and well-conducted. The paper is also written well, with an emphasis on good research practices. However, I have an issue with the proposed quicker/cheaper update for calculating the (diagonal empirical) Fisher Information Matrix for Online EWC ("Batch-EF"), as I detail later. If it were not for this, this paper would be a clear accept for me. I hope to resolve this issue with the authors during the discussion period, depending on which I can raise (or lower) my score.

**Pros of paper** (mostly already written in the "Review summary")

1. The paper is written well, with good detail and very good experiments.
2. The work is of significance for continual learning, with interesting conclusions.

**Cons of paper**

3. I am not convinced that the minibatching that the authors suggest (both for SI as an approximation to the AF and for EWC in the last paragraph of Section 5) is correct ("Batch-EF"). It appears to me that by minibatching instead of squaring each gradient element, one should obtain much worse approximations to the empirical Fisher information matrix ("EF").
Intuition: By calculating the gradient over minibatches and then squaring, one is reducing the noise that is being squared. Intuitively, this must affect the EF calculation in a bad way. For example, consider a full-batch calculation. In this case, Batch-EF will just be $g^2$. At the end of training, when we have converged to a low loss, this will be very small. In Appendix D.2, the authors argue that when gradient noise >> gradient, then Batch-EF $\approx$ EF, and derive that Batch-EF has larger values than EF. However, I expect Batch-EF to have smaller values than EF because of the reduced noise on average.
Additionally, I have myself experimented in the past with Batch-EF. I did not find that Batch-EF gave the same results as EF for EWC on similar benchmarks. I do not know why, in this paper, the authors found that the two gave same results (Table 1); perhaps it only works for specific hyperparameters.
Finally, a small note that may be of interest to the authors: the HAT codebase (github.com/joansj/hat) implements EWC as a baseline, however, my collaborators and I found that they implement EWC differently/incorrectly. One of the ways they are different is to do Batch-EF (along with other differences).
4. I am also not convinced that OnAF ("Online Absolute Fisher") and AF are / should be the same. After training, individual gradients should be relatively small (as we have converged to a solution), meaning that I would expect the AF to have small values in general. However, during training (especially near the beginning), gradients can be large, meaning that OnAF can end up having large values. Empirically, the Pearson correlations in Figure 2 (mid) show differences between the OnAF and AF versions.

**Additional suggestions to authors**

- The authors could consider adding a reference for the Absolute Fisher (Section 4.1). Can they say anything about the links between the Absolute Fisher and the empirical Fisher?
- Typo Section 4.2 second para: "Max-likeilhood"
- I felt that Section 5 got complicated, with many algorithms that need to be compared. I strongly recommend splitting the experiments part (Section 5.3) into experiments relevant for the two preceding sections (5.1 and 5.2) to reduce the complexity of writing.

**Update to review**

I am increasing my score from 5 to 6. I believe this paper is a good paper. However, an extremely extensive discussion with other reviewers has left some questions / concerns. Although I disagree with some of these, I agree with others:
- Some claims are overstated in the paper. The authors already changed these claims somewhat in the updated paper. Some reviewers are arguing for further changes. I think some claims can still be reduced, particularly, the link between AF and EF (and hence the link to EWC).
- One of the biggest reason I find this paper is interesting is not mentioned (enough) by the authors. In my opinion, this is a big reason why the work is significant, and if I were writing the paper, I would put it as one of the biggest motivations:
    -  There have been works recently looking at the Generalised Gauss-Newton approximation (= EF for classification), and trying to view optimisation algorithms as approximating the Hessian matrix. For example, see Khan et al., 2018 ("vAdam"), Kessler et al., 2020 ("BAdam"), Zhang et al., 2018 ("Noisy Adam"), Osawa et al., 2019 ("VOGN"). Such works provide evidence that different approximations of the EF can work well. Although I am not aware of previous works using the absolute value of gradients (as in AF), this paper provides evidence that such an approximation might be worth considering. Should we try and approximate the Fisher matrix in more ways in CL?
- Finally, it is my personal opinion (although others disagree) that the current paper is significant enough / provides enough insight already to be a good paper. However, performing a further state-of-the-art experiment or similar would undoubtedly improve the quality significantly.

I very much look forward to an updated version of this paper.

---

> ### Author Response · Authors · 2020-11-15
> **Response Reviewer 2**
>
> Thank you very much for your constructive, helpful feedback. We address your points by describing some additional experiments and by clarifying some points:
>
> **3. Mini-Batching of EF**
> We agree that this is counterintuitive at first. This is why we give the theoretical derivation in Appendix G2 (formerly D2); as Reviewer 4 pointed out, the relation observed in G.2 has also been studied elsewhere in the literature (see newly added refs in D2). Regarding the discrepancies between these findings and your intuition+experiments, see below.
>
> **Intuition vs Theory**
> The discrepancy between your intuition (Batch-EF should be smaller than EF) and our result likely stems from the following: We assume that each element of the minibatch is i.i.d. from the dataset, i.e. with replacement. This means that $E[\sigma_1\sigma_2]=0$ in the second line of the displayed equation in G.2. If mini-batch elements were drawn without replacement, $E[\sigma_1\sigma_2]$ would be a small, negative quantity. If we draw the full batch, the effect dominates and your intuition becomes true.
> As long as the minibatch is small, the effect should be negligible. We have added a section at the end of the paragraph to clarify.
>
> **Your experiments with Batch-EF**
> We have two hypotheses why you may have gotten different results for Batch-EF.
> -	In general, we found EWC to be somewhat unstable. Concretely, approximating the Fisher with more samples made performance of EWC more variable. We don’t fully understand why this is the case.
> The same applied to Batch-EF: more data made performance more unstable. To make comparisons fair, we used a fixed number of samples (one that worked well for EWC) and then gave the same number of samples to Batch-EF (which resulted in very few mini-batches, but similar performance to EWC).
> To sum up: In case your implementation of batch-EF used more mini-batches (higher total number of samples), this may explain why your observations differed.
> -	We also found that many small things (preprocessing, initialization, etc.) can affect different algorithms to different degrees.
>
>
>
> **4. OnAF and AF**
> We were also surprised by how similar AF and OnAF are, but the empirical findings seem to be unambiguous and robust. We directly check correlations of AF and OnAF in the appendix (J2, page 21) and they are large (0.8 on MNIST, 0.9 on CIFAR)
> It may be worth noting a few more things:
> -	We tried to make the formulation at the end of Sec 5.2 careful, since we agree that there could be settings where OnAF and AF are not as similar. We write “if OnAF and AF turn out to be similar, then it is clear that SI and AF are similar”. We also tried to point out that our confirmation of this claim is purely empirical and avoid claims that they are equal, but only similar/correlated.
> -	As we mention, the Adam-optimizer keeps an estimate of the second moment |g+sigma|^2 of the stochastic gradients. It does this with a very slowly moving average (beta_2=0.999, i.e. there’s non-negligible weight for gradients occurring 1000 iterations earlier, which is the same order of magnitude as the number of training iterations in our setup). While this is anecdotal evidence, it does suggest to us that |g+sigma|^2 and thus |g+sigma| (OnAF) changes quite slowly. A related remark: Using a slowly moving average (also of 0.999) for the sum, which SI accumulates for a task (equation 3), did slightly improve its performance on P-MNIST (by 0.3 percent, from 97.2 to 97.5, standard deviations <0.1).
> -	The intuition that gradients get much smaller at the end of training is not as true as one may think. This is visible in Fig 2, left and Fig G6, especially Tasks 4-6; the almost linear increase of SI suggests that the absolute values of gradients stay almost constant.
> That said, there is a short phase early in training with bigger gradients. We implemented a version of SI which puts no weight on this early phase and only accumulates its importances during the second half of training (epochs 10 to 20 of 20) and this did lead to better performance (also by 0.3 percent). So your intuition that the early, large gradients might harm SI (or its relation to AF) seems correct. The effect is fairly small.
>
> We have included the two improvements of SI, which put less weight on potentially harmful early gradients, in the Appendix and reference them in the discussion, since they demonstrate that understanding what’s going on allows improving algorithms. Thank you for pointing out your concerns, we think that this has helped improved our paper.
>
> Additional Suggestions:
> -	We could include a scatter/intensity plot showing how AF and (E)F are related empirically, but have no additional theoretical insights.
> -	Fixed the typo.
> -	Thanks for the constructive suggestion. We see your point and will think about how to make Section 5.3 easier to digest.
>
> Thanks again for your feedback - let us know if this addresses your points, or if we can include further clarifications or experiments.

---

> > ### Comment · AnonReviewer2 · 2020-11-18
> > **Response**
> >
> > Many thanks to the authors for addressing my comments directly. I intend to increase my score of this paper, after further discussion.
> >
> > **Batch-EF** My concerns about this method, although reduced, remain. I appreciate the weaker claims now made in the revised paper, as well as the more nuanced discussion in App G2.
> > - I can see that sampling with vs without replacement makes a difference. But this still does not sit right with me. Why does the theory require sampling with replacement while in practice, we only ever sample without replacement? Did the authors try comparing to sampling with replacement, and was that better performing?
> > - It seems quite concerning that EWC is unstable when using more samples for the Fisher. Does this mean that using the full-data Fisher is wrong, and we really need 'Fisher+noise' to do well? What implications does this have for all the algorithms that relate to the Fisher discussed in this paper? (This is not a criticism of the current paper, rather a potential way to improve its findings.)
> >
> > **OnAF vs AF** I am broadly happy with the response here. The updated manuscript looks much better here, thank you.
> > - The authors refer to Adam's $\beta_2$ parameter. Although 0.999 is indeed high, this does not mean that initial (absolute) gradients could theoretically still swamp out the actual AF (which would be calculated at only the converged value), as noted by the authors above. Therefore I think that using this as the main intuitive reason to say why OnAF and AF are similar (end of Sec 5.2) may be misleading. Indeed, my intuition would be that ignoring the first few epochs (or having a moving average), instead of the OnAF, would give better results. I appreciate the authors running these experiments. In my opinion, this would be good to discuss a little more in the main text.
> >
> > **AF vs EF** This seems to be a major point for other reviewers. For now, I note here that I agree that some claims made in the initial paper about AF being naturally related to EF should have been reduced, as they now have been in the main text. Perhaps a couple more sentences could be added discussing this to reduce any confusion for a reader. For me, the theoretical justification between the EF and AF, although desirable, is not necessary for this paper to be of significance to the community.

---

> > > ### Author Response · Authors · 2020-11-20
> > > **Re**
> > >
> > > Thanks for your reply, we appreciate your constructive feedback. It has lead us to understand at least one part of the experiments better.
> > > Let us know whether these changes and clarifications described below address your points.
> > >
> > > **Batch-EF**
> > >
> > > (1) The main reason for the assumption of sampling with replacement was that it makes the theoretical analysis cleaner. But we really think that for the small minibatch sizes used in our experiments, this 'not-quite-true' assumption has close to no effect:
> > > For example, if we sample 256 (our batch size) out of 60 000 MNIST images with replacement, then on average there will be ${\{256 \choose 2}}/{60 000} \approx 0.5$  pairs of identical images. So by replacing on average 0.5 images out of 256, we can turn the minibatch sampled with replacement into a minibatch without replacement. This is why sampling with and without replacement should be more or less identical and why we would not expect the difference in sampling strategy to have a measurable effect.
> > >
> > > (2) We looked more into the Fisher’s dependence on the number of samples and think we have gained a more satisfactory understanding: If we consider only one image and pass it through a fully connected ReLU network, there will be some `'dead' neurons with 0 activation. All the weights connected to dead neurons have gradients exactly equal to zero. Thus, the diagonal Fisher Information for these weights is also 0. The more images we consider, the less weights will have 0 Fisher Information - they just need to be 'active' for one image to be non-zero. So more samples lead to less weights with 0 importance, resulting in a less flexible network on latter tasks. Empirically, it seems quite plausible that this decreased flexibility leads to worse performance (of course there is a trade-off somewhere) and this would then explain our observation.
> > >
> > > We mention the effects of sample size in the paper now (end of Section 5), and give the explanation sketched above in Appendix C.1.1. There, we include empirical data showing that the number of weights with zero Fisher Information is large and decreases with the number of samples used for the Fisher (e.g. with 1000 samples, there are 2M weights with 0 importance; with 8000 samples, there are only 1.3M weights with 0 Fisher). We also discuss how this may be related to some of the optimal hyperparameter choices.
> > >
> > > It is worth noting that this (somewhat pathological) effect is not present for Conv-Nets, since there each weight receives inputs from many neurons. In fact, we found that the Fisher on CIFAR with the Conv-Net architecture, depended much less on the sample size than for the FC MNIST net (compare the two green lines in Figure J.2).
> > >
> > >
> > >
> > > **AF and OnAF**
> > > We have updated the parts of Sections 5.2 and 5.3, where we discuss AF and OnAF, to reflect this intuition and the evidence for it. We have removed some other remarks due to space limitations.
> > >
> > > **Absolute Fisher and real Fisher**
> > > We have updated the phrasing of Abstract, Intro and Discussion, also mentioning in the latter that AF has no clear theoretical interpretation beyond its similarity to the Fisher. We also dedicate a new section and figure to the relation between the two, as you had suggested originally and as was brought up by the AC.

---

### Author Response · Authors · 2020-11-15
**Updated, Improved Paper**

We would like to thank all reviewers for their input. We made several changes to the paper, including but not limited to new experiments. We invite the reviewers, especially no1 and no3 who had doubts about the usefulness of our paper, to read the updated Discussion, which includes experiments as well as predictions and improvements that can be made thanks to our framework. Of course, we also describe these and other changes in our detailed, individual replies to reviewers.

---

> ### Comment · Area_Chair1 · 2020-11-19
> **One unanswered concern is AF vs F**
>
> Thank you! I have read the changes. After reading the reviews, my understanding is that one (key) outstanding source of confusion among the reviewers (and myself) is the connection between AF and F. The paper is based on the assumption they are closely related. I wonder, are the two correlated in fact (in a single training, and across different runs)? If so, is it expected?

---

> > ### Author Response · Authors · 2020-11-20
> > **Empirical and Theoretical Similarity between AF and F**
> >
> > We completely missed this natural way to address the F-AF concerns, so thank you very much not only for having a look at paper&discussion, but also for this suggestion.
> >
> > Correlations are high, around 0.9 on Permuted-MNIST and around 0.85 on Split CIFAR (consistently across tasks and repetitions).
> >
> > Theoretically, this should be expected. The precise relation between AF and F depends on the gradient distribution, but if we - for example - assume that gradients are distributed normally $\mathcal{N}(\mu, \Sigma)$ with $\Sigma_{i,i} \gg \mu_i$ (corresponding to the observation that the noise is much bigger than the gradients), then we can see that $F \propto (AF)^2$. In fact this quadratic relation matches the empirical observations well (correlations between $(AF)^2$ and $F$ are always above 0.98 on CIFAR and between 0.86-0.96 on MNIST).
> >
> > We have added a new Section(6)&Figure(3) to the updated paper detailing the above. This should make the paper more complete and addresses AF-F concerns.

---

### Decision · Program_Chairs · 2021-01-07
**Final Decision**

**Decision:**

Reject

**Comment:**

The paper proposes a unification of three popular baseline regularizers in continual learning. The unification is realized through a claim that they all regularize (surprisingly) related objectives.

The key strengths of the paper highlighted by the reviewers were:
1. The established connection is valuable and interesting, even if weaker than suggested originally
2. Good motivation (unifying different regularization methods is useful for the community)
3. Clear writing

The key weakness of the paper is a weak empirical validation of the claim that these three regularizers work *because* they regularize the norm of the gradient (as mentioned in the discussion by R3). Rather, the key claims are correlational. The authors correctly say that (1) the three regularizers all regularize related objects (namely different norms of the gradient) and (2) they reduce forgetting. However, it is not sufficiently well demonstrated that (1) => (2). Relatedly, given that the paper does not have a very clear theoretical contribution, it would be really helpful to demonstrate utility. It would be useful to extend experiments that apply these insights to developing novel regularizers or improving/simplifying hyperparameter tuning.

Additionally, in the review process, the link was discovered to be weaker than originally suggested. The paper casts the relation in terms of the Fisher Information Matrix, which suggests it is theoretical and sound. After the discussion, it seems that viewing this relationship in terms of the Fisher Information Matrix is somewhat misleading. The three different regularization methods all regularize different norms of the gradient (L1 or L2), which are empirically, and under some assumption theoretically, related. More precisely, EWC regularizes the trace of the *Empirical* Fisher, which is equivalent to the L2 norm of the gradient of the loss function. SI regularizes a term similar to the L1 norm of the gradient. These effects were seen by the reviewers to be somewhat loosely related to the Fisher Information Matrix.

Based on the above, I have to recommend rejection. I would like to thank the Authors for submitting the work for consideration to ICLR. I hope the feedback will be useful for improving the work.